# VLN-MME: Diagnosing MLLMs as Language-guided Visual Navigation agents

## Abstract

Multimodal Large Language Models (MLLMs) have demonstrated remarkable capabilities across a wide range of vision-language tasks. However, their performance as embodied agents, which requires multi-round dialogue and sequential action prediction, needs further exploration. Our work investigates this potential in the context of Vision-and-Language Navigation (VLN) by introducing a unified and extensible evaluation framework to probe MLLMs as zero-shot agents by bridging traditional navigation datasets into a standardized benchmark, named VLN-MME. We simplify the evaluation with a highly modular and accessible design. This flexibility streamlines experiments, enabling structured comparisons and component-level ablations across diverse MLLM architectures, agent designs, and navigation tasks. Crucially, enabled by our framework, we observe that enhancing our baseline agent with Chain-of-Thought (CoT) reasoning and self-reflection leads to an unexpected performance decrease. This suggests MLLMs exhibit poor context awareness in embodied navigation tasks; although they can follow instructions and structure their output, their reasoning fidelity is low. VLN-MME lays the groundwork for systematic evaluation of general-purpose MLLMs in embodied navigation settings and reveals limitations in their sequential decision-making capabilities. We believe these findings offer crucial guidance for MLLM post-training as embodied agents.

## 1 Introduction

The rapid advancement of Multimodal Large Language Models (MLLMs) has raised interest in deploying them as embodied agents, moving beyond static vision-language tasks to dynamic, interactive decision-making. In this context, Vision-and-Language Navigation (VLN) (Anderson et al., 2018) emerges as a crucial and challenging paradigm to evaluate the MLLM's reasoning ability. Successfully navigating a 3D environment based on instructions requires more than pattern recognition; it fundamentally tests an agent's spatial understanding, its ability to plan and foresee the consequences of its actions, and its use of long-term memory to ground an extended plan. When navigation involves multi-round dialogue, it further probes the model's capacity for contextual reasoning. However, despite VLN's potential as a comprehensive benchmark for these core agentic skills, progress in systematically evaluating MLLMs is constrained by the limitations of existing evaluation pipelines.

First, embodied navigation tasks typically run in high-fidelity simulators such as Matterport3D (Chang et al., 2017) or Habitat (Savva et al., 2019). The evaluation cost grows sharply when large models are deployed as VLN agents in multi-round settings that require frequent interaction with the environment. Second, the existing VLN benchmarks are diverse (Anderson et al., 2018; Qi et al., 2020; Ku et al., 2020), and a single dataset can contain thousands of navigation trajectories, making comprehensive evaluation with large MLLM agents a prohibitively time-consuming and computationally heavy process. Third, prior studies often focus on improving success metrics with different LLMs, and rarely offer principled error analyses, which limits comparability and obscures the true contributions of model capability versus agent design.

More critically, recent approaches to evaluating MLLMs in VLN have gaps in understanding model behavior. On one hand, some works utilize end-to-end success metrics alone and are insufficient for understanding agent behavior. On the other hand, dedicated evaluation suites like NavBench (Qiao

et al., 2025), while comparing different models and tasks, do not systematically consider the crucial impact of varying agent designs. Consequently, the community still lacks a deeper understanding of how these models perform. Specifically, there is minimal fine-grained analysis of success and failure cases, error types, or patterns in agent decision-making. To address these limitations, we developed our own modular evaluation framework, designed specifically to diagnose MLLM behavior in navigation tasks. The necessity for such a framework is highlighted by a comparison with existing benchmarks in Table 1. Without the kind of diagnostic insights our approach provides, it is difficult to assess generalization, robustness, or the alignment between visual perception and instruction-following capabilities in MLLMs. As a result, progress in the field remains largely metric-driven, with little clarity on the underlying model behavior.

Table 1: Comparison of VLN benchmarks by key evaluation capabilities: support for diverse MLLMs and agent architectures, simulation-free execution, and fast evaluation.

| Benchmark | Diverse MLLM Support | Diverse Agent Support | Simulation Free | Evaluation Speed |
|---|---|---|---|---|
| R2R (2018) | ✗ | ✗ | ✗ | ✗ |
| VLNCE (2020) | ✗ | ✗ | ✗ | ✗ |
| NavBench (2025) | ✓ | ✗ | ✗ | ✓ |
| **Ours** | ✓ | ✓ | ✓ | ✓ |

In response to these gaps, we propose the **Vision Language Navigation Multi-Model Evaluation (VLN-MME)**, a novel evaluation framework designed to address these challenges head-on. Our approach is built on a modular and simulator-free architecture that prioritizes accessibility and reproducibility. Crucially, instead of focusing on high-level success metrics, we contribute a detailed error analysis that breaks down agent performance to evaluate core capabilities. This allows for a deeper understanding of an MLLM's proficiency in instruction following, spatial understanding, and historical sequential reasoning for long-horizon tasks.

Our contributions could be summarized as:

- We present a unified evaluation framework that enables structured, comparable assessment of different MLLMs, agents, and VLN tasks under a consistent interface.

- We introduce a simulator-free design that preserves navigational semantics while significantly reducing setup complexity and enabling broader accessibility.

- We curate and publish VLN data, environments, and configuration artifacts on public platforms to streamline benchmarking and reproducibility.

- We conduct an extensive and insightful error analysis that uncovers behavioral patterns and limitations in MLLMs' navigation reasoning.

This work aims to establish a standardized foundation for studying MLLMs in embodied environments, pushing the field beyond leaderboard metrics toward a deeper understanding of model behavior.

## 2 RELATED WORKS

**MLLMs as Embodied Navigation Agents**  The integration of Multimodal Large Language Models (MLLMs) into robotics has inspired new paradigms for Vision-and-Language Navigation (VLN). Early efforts leveraged LLMs to act as a copilot, providing high-level guidance to a specialist navigation agent (Qiao et al., 2023). More recently, work has explored using off-the-shelf MLLMs as zero-shot navigation agents through elaborate prompting (Zhou et al., 2024b), leading to more complex designs incorporating multi-agent collaboration (Long et al., 2023), topological maps (Chen et al., 2024), and self-evolving frameworks (Dong et al., 2025). Other works finetuning MLLMs on VLN data (Zhou et al., 2025; Lin et al., 2024; Pan et al., 2023; Zheng et al., 2023), adapting pre-trained video understanding models to navigation (Zhang et al., 2024b;a; Cheng et al., 2024; Zhang et al., 2025; Wei et al., 2025). However, the dynamic, iterative nature of embodied navigation makes the evaluation time-consuming and expensive. It hinders the scalable evaluation to understand agent behavior, calling for a flexible and representative evaluation pipeline.

**Evaluating MLLMs in Vision-Language Tasks**  Comprehensive evaluation benchmarks have emerged to test a wide spectrum of MLLM abilities (Chaoyou et al., 2023; Liu et al., 2024; Li et al., 2024b; Yue et al., 2024; Yu et al., 2024; Lu et al., 2023; Fei et al., 2025), from perception to cognition. However, the evaluation paradigm for these benchmarks is overwhelmingly centered on static, single-turn tasks, where a model provides a single response to a given visual-textual input.

Consequently, while these benchmarks can measure an MLLM's ability to make an isolated correct judgment, they do not capture its capacity for the sustained, sequential reasoning essential for executing a successful multi-step plan. The most similar work to us is NavBench (Qiao et al., 2025) However, its analysis is limited to a single, pre-defined agent formulation, precluding any comparison of different agent strategies or designs. Furthermore, most evaluation frameworks (Zhou et al., 2024b; Chen et al., 2024) focus on reporting aggregate performance, lacking the detailed, episode-level error analysis necessary to diagnose precisely why an agent succeeds or fails. To the best of our knowledge, no existing work provides a unified framework for evaluating MLLMs in navigation that jointly considers a variety of agent strategies, MLLM architectures, and datasets. Our work is designed to fill this critical gap, enabling a deeper, more systematic analysis of MLLM-based navigation agents.

## 3 METHOD

### 3.1 A MODULAR FRAMEWORK FOR VLN EVALUATION

To enable systematic and reproducible research on MLLMs in embodied settings, we designed and implemented a modular software stack for VLN evaluation. Our architecture enforces a clean separation of concerns between its primary components: the model, the agent, and the environment. This modularity empowers us to seamlessly interchange different MLLMs, implement novel agent designs, or introduce new datasets for structured comparisons and component-level ablations. The high-level architecture of our framework is illustrated in Figure 1.

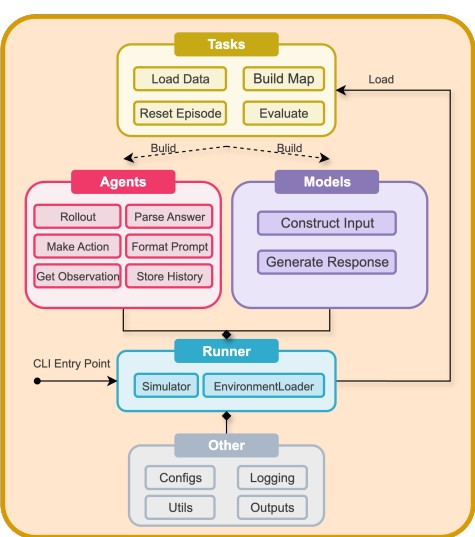

Figure 1: A high-level structure for the benchmark, centered on the interplay between **Tasks**, **Agents**, and **Models**.

Our framework is built upon three primary components: **Model**, **Agent**, and **Dataset**, to enable evaluation across both model and agent design axes. The **Model** component serves as an abstraction layer, providing a unified interface to support a wide variety of MLLMs by handling model-specific API calls. The **Agent** is the core decision-making module that mediates the interaction between the MLLM and the environment. Its primary responsibility is to translate the current environmental state, including visual observations and navigable options, into a structured prompt for the MLLM. Subsequently, it parses the model's textual output to derive an executable action and interact with the environment. In VLN-MME, we distinguish agent designs by their memory mechanism, and we implement agents that maintain a natural language description of past instructions and observations as our baselines. Moreover, we implement enhanced variations of baselines that integrate reasoning strategies in agent design, such as chain-of-thought (CoT) prompting (Wei et al., 2022) and post-action reflection (Yao et al., 2022).

At each decision step, the MLLM receives a rich, multimodal prompt. The visual input is a panoramic image of the agent's surroundings, with navigable viewpoints annotated by numerical markers. The textual component is structured to provide context progressively: it begins with a system prompt defining the task rules and the specific navigation instruction. For agents using a text map as memory (Chen et al., 2024), the global connectivity of their discovered symbolic map is provided next. The prompt then includes the agent's history, which differs based on the memory mechanism. For

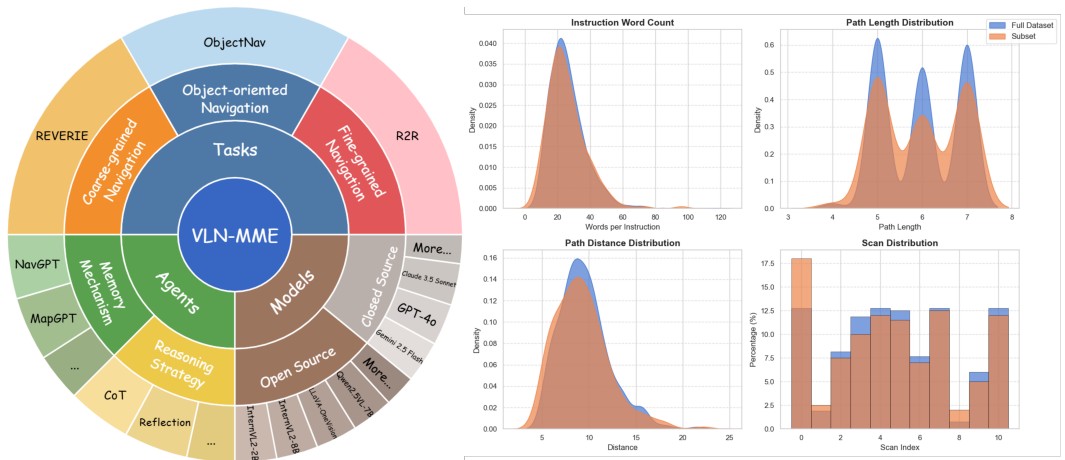

Figure 2: Overview of the VLN-MME benchmark. **(Left)** The composition of the benchmark, detailing the diverse set of **Tasks**, **Agents**, and **Models** it supports. **(Right)** A statistical comparison of our benchmark's R2R data subset against the original R2R val_unseen split, showing similar distributions for key metrics like instruction word count and path length.

agents relying on text summarization as memory (Zhou et al., 2024b), this history consists of a simple sequence of prior actions. In contrast, for agents employing a text map, the history is more comprehensive, augmented at each step with the scene summary of the current node and lists of visited and unvisited nodes. Following the history, the agent's current heading and elevation are specified. The prompt concludes with a structured dictionary of available actions, which organizes navigable options by their relative direction, mapping each candidate marker to its caption.

To ensure modularity and ease of extension, we employ a unified **factory pattern** for instantiating all three component types. Each component is associated with a unique string identifier in a central registry. At runtime, a dynamic loader uses this identifier to import and construct the desired class. This design enables true "plug-and-play" capability; integrating a new agent, for instance, simply requires adding its class to the agents directory and an entry to the registry, with no changes to the core evaluation logic.

The orchestration of these components is managed by a central **Runner** module, which uses an efficient configuration system for easy and reproducible experiment setup. The Runner handles the entire evaluation lifecycle. It begins by loading the pre-stored simulator-free environment, whose construction is detailed in Section 3.3. Concurrently, it dynamically loads the specified dataset and splits via the factory, as described in Section 3.2. During an episode, the Runner acts as the low-level intermediary between the agent and the environment; it services agent requests for state information, renders observations, and executes actions. Throughout this process, the Runner logs all interactions for detailed post-hoc analysis. Upon completion of all episodes, it is responsible for calculating and reporting the final evaluation metrics. This centralized design cleanly separates high-level agent logic from low-level environment management, reinforcing the framework's modularity.

## 3.2 DATASET CONSTRUCTION FOR EFFICIENT EVALUATION

To address the computational challenges of evaluating large models on existing, large-scale VLN datasets and to facilitate rapid experimentation, we constructed a curated benchmark for efficient yet representative evaluation. Follow the broader definiation of VLN (Zheng et al., 2023; Zhou et al., 2024a), our benchmark is composed of samples carefully drawn from the validation unseen splits of three main datasets: R2R (Anderson et al., 2018), REVERIE (Qi et al., 2020), and ObjectNav (Batra et al., 2020). The primary goal is to offer a lightweight benchmark that significantly reduces evaluation overhead while faithfully preserving the distributional characteristics of the full benchmarks. This ensures our benchmark can serve as a reliable proxy, allowing for efficient validation that aligns with previous evaluation methods.

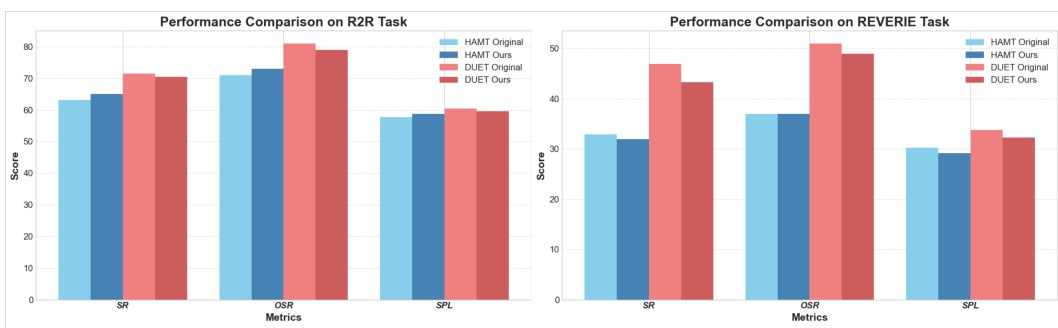

Figure 3: Comparison of model performance on full val_unseen splits vs. our curated benchmark for R2R and REVERIE.

Our construction strategy employs a task-specific stratified sampling process designed to maintain diversity across three key axes: **scene complexity**, **path difficulty**, and **linguistic richness**. For instance, when constructing the R2R portion of our benchmark from the original 783 unique trajectories, the process begins by stratifying episodes based on their Matterport3D scan ID to ensure the selection reflects the original distribution of environments. Within each scan-based group, trajectories are then binned by their path length - a proxy for navigational difficulty - and sampled proportionally from each bin. Finally, to ensure linguistic variety, one of the three available natural language instructions is selected at random for each chosen trajectory. A similar stratified methodology, adapted to the unique characteristics of each task, was applied to create the benchmark data for REVERIE. For ObjectNav, we also consider object balance, ensuring that the sampled object navigation episodes maintain a balanced distribution of objects from the previous benchmark. This meticulous process ensures that our resulting benchmark, while significantly smaller, retains a comparable distribution of these core characteristics to the original datasets, as illustrated in Figure 2.

To validate the fidelity of our constructed benchmark, we evaluated three high-performing specialist VLN agents, HAMT (Chen et al., 2021), and DUET (Chen et al., 2022), on both the full val_unseen splits and our curated benchmark for R2R and REVERIE. The results, presented in Figure 3, reveal a strong correlation in performance. Key metrics such as Success Rate (SR) and Success weighted by Path Length (SPL) on our benchmark closely track the performance on the full splits, with deviations typically within a 2-3 percentage point margin. This close alignment confirms that our stratified sampling approach successfully captures the intrinsic difficulty and diversity of the original datasets, establishing our benchmark as a reliable and efficient proxy for full-scale MLLM evaluation.

### 3.3 SIMULATOR-FREE ENVIRONMENT DESIGN

While powerful, relying on simulators for real-time rendering at each step introduces a significant computational bottleneck, especially when evaluating large models at scale across numerous tasks and agent designs. To address this challenge and maximize accessibility, our framework introduces a **simulator-free** mode. This is achieved by pre-rendering and storing all necessary visual observations and environmental metadata, enabling lightweight and highly scalable execution of navigation tasks.

The core of this mode is a pre-rendered panoramic observation set for each viewpoint in the environment. Instead of real-time rendering, we capture a set of four non-overlapping perspective images at each location, each with a 90° Field of View (FOV), which together form a complete 360° visual context. Crucially, all navigable directions are annotated directly onto these images using visually distinct numerical markers. These numbers reflect the ordering of navigable candidates based on their global heading angles, derived from the navigation graph. For example, neighbors are sorted by increasing global angle relative to the current orientation, and the assigned marker numbers (e.g., 1, 2, 3) follow this order.

To enhance model understanding, each marked neighbor viewpoint is also annotated with a caption generated by GPT-4o. To generate these, GPT-4o was prompted to describe the scene visible at

the marked location and what navigating towards it would likely reveal (e.g., "A hallway leading to a bright living room"). Additionally, each viewpoint is summarized with a GPT-4o-generated scene description, providing global context for map-based agents. All visual and semantic assets are published on open-source platforms and are managed directly by our framework, which handles automatic downloading for ease of use. This includes all environmental information such as the pre-rendered panoramic images, connectivity data between viewpoints, and precomputed graph utilities like shortest-path geodesic distances for efficient metric calculation. We provide complete task splits for all dataset in this simulator-free format to ensure immediate accessibility.

## 4 EXPERIMENTS

### 4.1 SETTINGS

**Evaluation Metrics.**   In this work, we focus exclusively on the navigation component of both R2R and REVERIE tasks, without considering object grounding in REVERIE. We adopt a standard set of navigation metrics to evaluate agent performance: (1) *Trajectory Length* (TL), which measures the average path length in meters; (2) *Navigation Error* (NE), the average distance between the agent's final position and the goal location; (3) *Success Rate* (SR), the percentage of episodes where the final location is within 3 meters of the target; (4) *Oracle Success Rate* (OSR), the success rate assuming an optimal stopping policy; (5) *Success weighted by Path Length* (SPL) (Jain et al., 2019), which combines success with path efficiency; (6) *Normalized Dynamic Time Warping* (nDTW) (Ilharco et al., 2019), which measures the trajectory similarity to the ground truth path; and (7) *Success weighted by normalized DTW* (SDTW), a combined metric capturing both goal-reaching and trajectory fidelity.

**Implementation Details**   We evaluate four open-source Multimodal Large Language Models (MLLMs) in a zero-shot setting: Qwen2.5-VL-7B (Bai et al., 2025), InternVL3-2/8B (Zhu et al., 2025), LLaVA-One-Vision-7B (Li et al., 2024a). These models are integrated into eight distinct agent configurations, categorized into two primary classes: agents using text summarization as memory and agents using a text map as memory. Each class includes four variants: a baseline, one with Chain-of-Thought (CoT) prompting, one with reflection-based reasoning, and one featuring both CoT and reflection. To ensure efficient inference and memory management for these large models, all agents are served using the vLLM backend (Kwon et al., 2023). We assess their performance on all the tasks in our benchmark, additionally, we compare these zero-shot agents against previously finetuned Vision-Language Model (VLM) agents and finetuned MLLM agents on the R2R and REVERIE tasks, evaluating performance across both the full dataset from prior evaluation methods and our benchmark. All experiments are conducted on a single NVIDIA A100 GPU with 40GB VRAM.

### 4.2 PERFORMANCE

We evaluate our zero-shot MLLM-based agents and compare their performance against prior state-of-the-art finetuned agents. Our analysis is structured around two key comparisons: first, a macro-level comparison against finetuned methods to contextualize the zero-shot paradigm, and second, a micro-level analysis of the different MLLMs, agent architectures, and reasoning strategies.

Our main results, detailed in Table 2 and illustrated in Figure 4, offer insights into the performance of different MLLMs, agent architectures, and reasoning techniques in a zero-shot setting. Among the evaluated MLLMs, Qwen2.5-VL-7B consistently emerges as the most capable navigation agent, as demonstrated in the 3D bar chart comparing text-summarization memory-based agent variants. It achieves the highest success rates across the majority of tasks, with InternVL3-8B also showing decent performance capabilities. For example, in the baseline NavGPT configuration on the fine-grained R2R task, Qwen2.5-VL-7B obtains a success rate of 27.5%, substantially outperforming LLaVA-OneVision (11.5%) and InternVL3-2B (13.5%).

Surprisingly, it is counterintuitive that the integration of advanced prompting strategies like Chain-of-Thought (CoT) and reflection does not consistently yield performance improvements and can be detrimental. For instance, on the fine-grained navigation task (Table 2), applying CoT and reflection to the Qwen-2.5-VL-7B model decreases its Success Rate (SR) by 5.5% and 2.0%, respectively. This is not an isolated case, as the performance degradation is a consistent trend across all evaluated

| Agent / MLLM | Fine-Grained Navigation | | | | | | | | Coarse-grained Navigation | | | | | | | | Object-Oriented Navigation | | | | | | | |
|---|---|---|---|---|---|---|---|---|---|---|---|---|---|---|---|---|---|---|---|---|---|---|---|---|
| | TL | NE | SR | OSR | SPL | nDTW | SDTW | CLS | TL | NE | SR | OSR | SPL | nDTW | SDTW | CLS | TL | NE | SR | OSR | SPL | nDTW | SDTW | CLS |
| **Text Summarization as Memory** | | | | | | | | | | | | | | | | | | | | | | | | |
| **NavGPT** | | | | | | | | | | | | | | | | | | | | | | | | |
| InternVL3-2B | 9.89 | 8.56 | 13.50 | 27.00 | 5.46 | 21.25 | 5.75 | 21.59 | 10.13 | 10.18 | 7.33 | 16.33 | 2.50 | 15.30 | 2.97 | 17.17 | 10.35 | 6.27 | 21.50 | 40.50 | 3.57 | 13.67 | 2.55 | 15.57 |
| InternVL3-8B | 11.74 | 7.55 | **28.00** | **50.50** | 12.61 | 25.28 | 13.38 | 26.22 | 11.90 | 9.25 | **20.00** | **30.67** | 7.18 | 17.32 | 8.18 | 18.84 | 11.55 | **4.63** | 39.00 | 56.00 | 7.69 | 14.51 | 5.09 | 17.36 |
| LLaVA-OV-7B | 8.04 | 8.40 | 11.50 | 20.50 | 4.94 | 25.28 | 5.70 | 25.53 | 9.85 | 9.35 | 14.67 | 20.00 | 5.19 | 19.27 | 6.09 | 18.39 | 9.52 | 5.93 | 27.50 | 41.00 | 4.51 | 16.54 | 5.11 | 14.71 |
| Qwen2.5-VL-7B | 8.54 | **6.99** | 27.50 | 44.00 | **17.11** | **35.97** | **18.88** | **34.85** | 8.94 | **8.55** | 18.67 | 27.33 | **9.00** | **23.79** | **9.88** | **23.55** | **9.07** | 4.65 | 37.50 | **56.50** | **13.18** | **21.83** | **10.63** | **23.52** |
| **NavGPT w/ CoT** | | | | | | | | | | | | | | | | | | | | | | | | |
| InternVL3-2B | **6.21** | 8.84 | 8.00 | 14.50 | 4.47 | 26.15 | 4.28 | 27.45 | **4.98** | 9.87 | 5.33 | 9.00 | 3.24 | 23.39 | 3.11 | 25.09 | 6.78 | 5.56 | 25.00 | 41.00 | 6.66 | 25.51 | 5.34 | 23.98 |
| InternVL3-8B | 9.07 | **7.56** | 19.00 | 35.50 | 10.95 | 29.22 | 10.90 | 29.59 | 7.96 | **9.39** | 15.33 | 22.00 | **9.31** | **24.93** | **9.14** | **26.40** | 6.43 | 5.31 | **34.50** | 43.50 | 12.67 | 27.22 | 10.30 | 27.35 |
| LLaVA-OV-7B | 7.97 | 8.85 | 12.50 | 22.50 | 5.41 | 23.20 | 5.60 | 24.43 | 8.43 | 9.47 | 14.00 | 20.33 | 5.90 | 21.09 | 7.13 | 21.38 | 9.69 | 5.66 | 33.50 | **47.50** | 7.31 | 15.90 | 6.24 | 17.92 |
| Qwen2.5-VL-7B | 9.04 | 7.97 | **21.00** | **37.50** | **11.41** | **30.23** | **11.56** | **31.36** | 8.29 | 9.85 | **15.67** | **24.33** | 8.10 | 24.02 | 8.82 | 25.59 | 6.86 | **5.13** | 33.00 | 44.50 | **13.25** | **28.63** | **12.54** | **28.32** |
| **NavGPT w/ Reflection** | | | | | | | | | | | | | | | | | | | | | | | | |
| InternVL3-2B | 6.50 | 8.94 | 8.00 | 16.00 | 5.20 | 29.91 | 5.00 | 30.13 | 7.20 | 9.41 | 8.50 | 15.00 | 4.80 | 25.49 | 4.60 | 24.93 | **6.43** | 5.56 | 28.00 | 43.00 | 7.83 | **26.53** | 6.14 | **25.28** |
| InternVL3-8B | 4.54 | 7.99 | 12.00 | 19.00 | 9.53 | 34.25 | 8.89 | 34.44 | 6.82 | 10.20 | 11.00 | 17.33 | 5.97 | 23.58 | 6.04 | 27.79 | 8.35 | **5.02** | 32.50 | **51.00** | 9.12 | 18.79 | 6.68 | 22.14 |
| LLaVA-OV-7B | **2.81** | 8.01 | 10.50 | 11.00 | 9.44 | **38.17** | 8.43 | **38.39** | **5.15** | 9.34 | 9.33 | 14.67 | 5.82 | **28.04** | 5.91 | **30.22** | 9.35 | 5.58 | 34.00 | 47.50 | 7.90 | 15.48 | 6.11 | 17.05 |
| Qwen2.5-VL-7B | 6.93 | **7.17** | **24.00** | 32.50 | **14.95** | 36.51 | **16.44** | 33.59 | 6.96 | **8.76** | **12.00** | **18.00** | **7.97** | 26.88 | **7.97** | 25.78 | 7.55 | 5.06 | **35.50** | 50.50 | **14.67** | 23.33 | **11.04** | 25.23 |
| **NavGPT w/ CoT & Reflection** | | | | | | | | | | | | | | | | | | | | | | | | |
| InternVL3-2B | **7.15** | 9.24 | 4.50 | 15.00 | 1.70 | 22.45 | 2.05 | 23.47 | 7.30 | 9.78 | 9.33 | 15.00 | 4.63 | 22.33 | 5.30 | 24.07 | **6.94** | 6.25 | 24.50 | 37.50 | 7.26 | 21.29 | 6.11 | 21.47 |
| InternVL3-8B | 7.22 | **7.47** | 22.00 | 32.50 | 15.33 | **36.62** | 16.18 | **35.98** | 8.95 | 9.07 | **17.33** | **28.67** | **10.07** | 24.11 | **10.24** | 25.78 | 9.18 | **5.30** | 32.50 | **51.50** | 8.14 | 18.59 | 5.99 | 21.45 |
| LLaVA-OV-7B | 7.61 | 8.48 | 10.00 | 22.00 | 5.83 | 28.01 | 6.31 | 26.32 | 8.44 | **8.68** | 14.00 | 22.00 | 6.78 | 24.73 | 8.37 | 22.60 | 8.55 | 5.66 | 28.50 | 44.00 | 7.25 | 21.26 | 6.74 | 19.69 |
| Qwen2.5-VL-7B | 7.82 | 7.53 | **25.50** | **38.50** | **17.68** | 34.86 | **17.65** | 34.80 | 7.19 | 9.48 | 11.67 | 18.00 | 7.89 | **27.07** | 7.81 | **28.22** | 7.60 | 5.39 | **36.00** | 47.00 | **13.67** | **26.52** | **11.49** | **26.98** |
| **Text Map Memory Agents** | | | | | | | | | | | | | | | | | | | | | | | | |
| **MapGPT** | | | | | | | | | | | | | | | | | | | | | | | | |
| InternVL3-2B | 9.84 | 8.61 | 11.00 | 22.50 | 3.71 | 20.18 | 3.85 | 21.89 | 10.19 | 9.59 | 12.00 | 19.00 | 4.41 | 18.10 | 5.24 | 19.97 | 10.35 | 5.93 | 27.50 | 46.50 | 4.41 | 13.59 | 3.69 | 15.57 |
| InternVL3-8B | 6.78 | 7.70 | 18.00 | 32.00 | 12.46 | **34.34** | 13.06 | **33.78** | 7.26 | 9.16 | 13.67 | 22.33 | 7.87 | **26.63** | 8.04 | **27.62** | 5.95 | 5.26 | 31.50 | 44.50 | **11.61** | **28.03** | 8.98 | **27.39** |
| LLaVA-OV-7B | 4.97 | 8.44 | 8.50 | 15.50 | 5.59 | 31.70 | 5.29 | 32.62 | 8.58 | 8.96 | 14.67 | 22.00 | 6.48 | 23.33 | 7.33 | 22.39 | 7.92 | 5.86 | 22.50 | 35.50 | 4.28 | 20.66 | 4.63 | 18.34 |
| Qwen2.5-VL-7B | 8.16 | **7.13** | **26.00** | **38.00** | **17.31** | 34.31 | **17.39** | 33.37 | 10.52 | **8.53** | **21.67** | **32.33** | **8.96** | 21.27 | **10.85** | 20.78 | 9.77 | **4.82** | **36.50** | **52.50** | 11.05 | 20.06 | **9.06** | 22.13 |
| **MapGPT w/ CoT** | | | | | | | | | | | | | | | | | | | | | | | | |
| InternVL3-2B | **5.09** | 8.99 | 6.50 | 14.00 | 4.12 | 26.43 | 3.94 | 27.62 | **5.22** | 9.88 | 4.00 | 8.00 | 1.58 | 23.01 | 1.77 | 24.48 | 6.66 | 6.31 | 20.00 | 35.50 | 4.76 | 21.14 | 4.69 | 21.05 |
| InternVL3-8B | 5.96 | 8.79 | 12.00 | 22.50 | 9.66 | **31.15** | 8.64 | **33.11** | 5.82 | 9.07 | 13.33 | 17.33 | 8.77 | 30.01 | 9.09 | 30.77 | 6.55 | 5.34 | **34.00** | 46.50 | **12.33** | 26.74 | **9.38** | 27.22 |
| LLaVA-OV-7B | 8.34 | 8.96 | 8.50 | 13.50 | 2.83 | 19.61 | 2.84 | 19.10 | 7.82 | 9.63 | 8.67 | 13.00 | 3.38 | 20.22 | 4.10 | 21.21 | 7.64 | 6.43 | 16.00 | 31.00 | 4.14 | 19.11 | 3.21 | 18.63 |
| Qwen2.5-VL-7B | 7.45 | **7.78** | **17.00** | **28.50** | **10.47** | 30.38 | **10.86** | 29.67 | 6.48 | **8.49** | **16.33** | **21.00** | **10.61** | **30.81** | **11.11** | **31.34** | 9.51 | **4.78** | 32.00 | **54.00** | 9.95 | 22.14 | 8.65 | 23.29 |
| **MapGPT w/ Reflection** | | | | | | | | | | | | | | | | | | | | | | | | |
| InternVL3-2B | **2.37** | 8.55 | 4.00 | 6.00 | 3.26 | 32.83 | 2.97 | 33.54 | **2.45** | 9.58 | 3.67 | 5.00 | 3.31 | 27.97 | 2.81 | **31.14** | 9.72 | 5.85 | 25.00 | 44.00 | 4.50 | 15.52 | 3.04 | 17.39 |
| InternVL3-8B | 5.85 | 7.80 | 16.50 | 28.50 | **10.89** | **36.05** | 11.48 | **35.18** | 6.49 | 8.80 | 12.67 | 19.33 | **6.64** | 27.19 | 6.91 | 28.24 | **6.23** | 5.50 | 30.00 | 40.50 | **10.36** | **26.92** | **8.12** | **25.74** |
| LLaVA-OV-7B | 8.35 | 8.50 | 10.00 | 20.00 | 5.50 | 26.62 | 5.50 | 23.33 | 8.20 | 9.47 | 11.00 | 19.00 | 6.00 | 25.84 | 6.00 | 25.16 | 9.46 | 6.11 | 15.50 | 33.50 | 2.02 | 15.90 | 2.53 | 15.57 |
| Qwen2.5-VL-7B | 10.41 | **7.12** | **26.50** | **41.00** | 10.12 | 27.97 | **12.88** | 25.38 | 9.60 | **8.67** | **15.67** | **23.67** | 6.00 | 23.29 | **7.62** | 20.63 | 10.15 | **4.90** | **33.50** | **46.00** | 7.41 | 17.18 | 6.33 | 17.34 |
| **MapGPT w/ CoT & Reflection** | | | | | | | | | | | | | | | | | | | | | | | | |
| InternVL3-2B | 7.52 | 8.82 | 9.00 | 20.00 | 4.88 | 23.45 | 4.98 | 24.08 | 7.35 | 9.95 | 4.67 | 12.67 | 1.61 | 19.60 | 2.02 | 21.94 | **7.26** | 6.15 | 18.00 | 35.50 | 4.74 | **22.73** | 3.77 | **22.14** |
| InternVL3-8B | 8.54 | **8.42** | **18.00** | **34.50** | **10.84** | **28.16** | **10.84** | **29.42** | 9.64 | 9.76 | **13.00** | **24.00** | 6.06 | 20.36 | 6.75 | 22.16 | 8.95 | **5.38** | **33.50** | **53.50** | **8.01** | 16.44 | **5.85** | 19.04 |
| LLaVA-OV-7B | 8.35 | 8.50 | 13.00 | 23.00 | 5.81 | 25.55 | 6.04 | 25.47 | 8.14 | 9.56 | 9.00 | 17.67 | 4.53 | 21.74 | 4.86 | 21.98 | 8.53 | 5.74 | 24.50 | 37.00 | 5.45 | 21.01 | 4.35 | 20.49 |
| Qwen2.5-VL-7B | 6.29 | 9.07 | 14.00 | 20.50 | 7.12 | 25.82 | 7.03 | 25.41 | **7.09** | **8.94** | 10.67 | 18.67 | **6.68** | **25.66** | **7.27** | **25.81** | 7.45 | 5.90 | 25.50 | 36.50 | 6.78 | 22.67 | 5.23 | 21.56 |

Table 2: Performance Comparison of MLLM-based Agents on VLN-MME. Agents are grouped by their primary architecture type. Best performance per group is marked in bold.

| Methods | R2R | | | | | | | | | | REVERIE | | | | | | | | | |
|---|---|---|---|---|---|---|---|---|---|---|---|---|---|---|---|---|---|---|---|---|
| | Val Unseen | | | | | Subset | | | | | Val Unseen | | | | | Subset | | | | |
| | TL | NE | OSR | SR | SPL | TL | NE | OSR | SR | SPL | TL | NE | OSR | SR | SPL | TL | NE | OSR | SR | SPL |
| *VLN Specialist*: | | | | | | | | | | | | | | | | | | | | |
| VLNBERT (2021) | 12.01 | 3.93 | 70 | 63 | 57 | 12.01 | 3.76 | 70 | 63 | 56 | - | - | - | - | - | - | - | - | - | - |
| HAMT (2021) | 12.14 | 3.92 | 71 | 63 | 58 | 12.10 | 3.51 | 73 | 65 | 59 | 14.08 | - | 37 | 33 | 30 | 14.62 | - | 37 | 32 | 29 |
| DUET (2022) | 13.94 | 3.31 | 81 | 72 | 60 | 13.25 | 3.54 | 79 | 71 | 60 | 22.11 | - | 51 | 47 | 34 | 22.28 | - | 49 | 43 | 32 |
| *Finetuned MLLM*: | | | | | | | | | | | | | | | | | | | | |
| NaviLLM (2023) | 12.81 | 3.38 | 81 | 66 | 54 | 15.80 | 3.32 | 86 | 66 | 55 | 16.04 | 5.76 | 54 | 45 | 37 | 19.02 | 5.80 | 55 | 34 | 27 |
| NavGPT-2 (2025) | 12.79 | 3.35 | 78 | 70 | 67 | 12.39 | 3.04 | 82 | 74 | 62 | - | - | - | - | - | - | - | - | - | - |
| *Zero-shot MLLM*: | | | | | | | | | | | | | | | | | | | | |
| InternVL3-2B | 20.69 | 9.19 | 21 | 9 | 3 | 20.36 | 8.56 | 27 | 14 | 5 | 21.57 | 10.10 | 17 | 9 | 4 | 21.71 | 10.18 | 16 | 7 | 3 |
| InternVL3-8B | 23.23 | 7.97 | 42 | 21 | 9 | 23.77 | 7.55 | 51 | 28 | 13 | 24.45 | 8.85 | 29 | 19 | 7 | 24.52 | 9.25 | 31 | 20 | 7 |
| LLaVA-OV-7B | 16.87 | 8.22 | 22 | 12 | 6 | 16.58 | 8.40 | 21 | 12 | 5 | 21.08 | 9.09 | 22 | 14 | 6 | 20.86 | 9.35 | 20 | 15 | 5 |
| Qwen2.5-VL | 14.89 | 7.63 | 32 | 20 | 13 | 17.20 | 6.99 | 44 | 28 | 17 | 17.59 | 8.75 | 23 | 16 | 7 | 18.28 | 8.55 | 27 | 19 | 9 |

Table 3: Performance of baseline agents on the R2R and REVERIE tasks, with results compared across the previous and our benchmark.

MLLMs and tasks. While applying CoT or reflection individually often reduces performance, their combination (CoT+Reflection) occasionally outperforms using either technique alone.

Moreover, the choice between architectures using text summarization as memory versus those using a text map as memory does not yield a universally superior agent, with performance being highly model-dependent. As shown in Table 2, using a text map as memory provides benefits for certain

models on specific tasks. For example, smaller MLLMs like InternVL3-2B gain a slight boost in success rate on the Coarse-grained navigation task. However, the opposite pattern emerges for others, indicating that architectural preferences vary significantly across different MLLMs.

We argue that the primary issue is the model's poor context awareness when situated in an embodied navigation task. We investigate this hypothesis by analyzing the logical coherence of the model's Chain-of-Thought reasoning and its self-reflection. This examination reveals two key, interrelated flaws. First, the model exhibits a strong tendency towards 'local' reasoning, where its decisions are driven almost exclusively by the immediate visual input, largely neglecting the rich context provided by its action and observation history. Second, as a direct result of this limited historical perspective, the model struggles to understand the downstream consequences of its actions, failing to adapt its strategy or recover from errors in the long-term, sequential flow of the task. A comprehensive error analysis in Section 4.3 provides further evidence to support this conclusion.

As shown in Table 3, a significant performance gap exists between our zero-shot MLLM agents and prior finetuned agents on both R2R and REVERIE. For instance, on the R2R Val Unseen split, the best finetuned agents like DUET achieve a Success Rate (SR) of 72%, whereas our best-performing zero-shot agent, Qwen2.5-VL-7B, reaches 18% SR. This highlights the inherent challenge of zero-shot navigation and the effectiveness of task-specific training. Nevertheless, the zero-shot agents demonstrate promising, non-trivial navigation capabilities, establishing a crucial baseline for this emerging paradigm. We also observe that the performance of prior methods on our benchmark subset is largely consistent with their results on the full validation set, further validating the representativeness of our subset for evaluation.

The results reveal clear difficulty hierarchies across navigation tasks. Object-Oriented Navigation proves most tractable, with agents consistently achieving the highest success rates (up to 39.0%). Fine-grained navigation presents moderate difficulty, while coarse-grained navigation emerges as the most challenging task, with substantially lower success rates across all models. This suggests that navigating to less precisely defined locations based on high-level instructions represents a particularly difficult challenge for current zero-shot MLLMs.

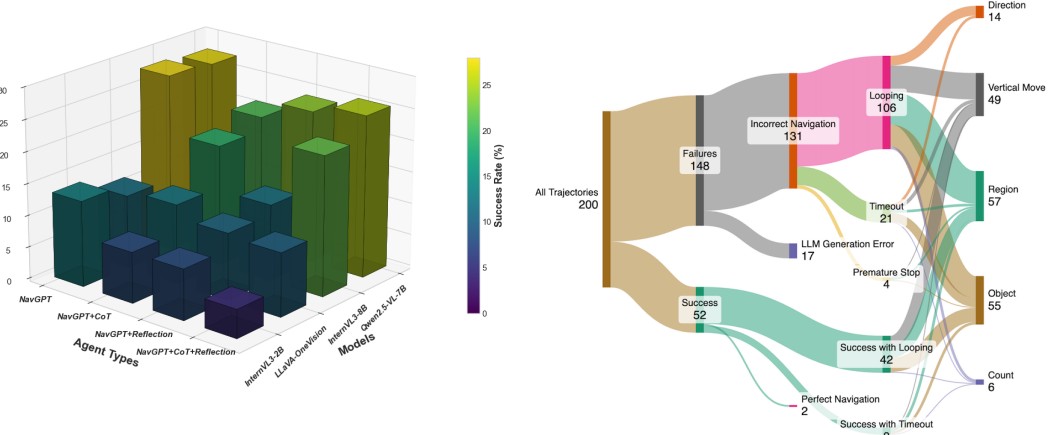

Figure 4: Performance comparison of agents using text summarization memory under different reasoning strategies across multiple backbone MLLMs.

Figure 5: A high-level analysis of success and failure modes for Qwen2.5-VL-7B model using an agent with text map memory.

## 4.3 DISCUSSION

As discussed in section 4.2, we reveal some counterfactual behavior when MLLMs performing embodied navigation. We further conduct an error analysis to understand their error pattern and find that they are hindered by fundamental limitations across several cognitive dimensions. Interestingly,

we find that the high navigation failure rate is overwhelmingly dominated by looping behaviors, shown in Figure 5. It is not a superficial issue but symptomatic of deeper challenges in instruction fidelity, spatial reasoning, historical context utilization, and the grounding of multimodal perception into action. We discuss these three interconnected aspects below.

**Instruction Following and Reasoning Fidelity.** A primary challenge is the limited fidelity with which MLLMs adhere to complex instructions, particularly those governing their reasoning process. While the models can follow basic output formatting prompts, they struggle with more abstract meta-instructions. For instance, when prompted with Chain-of-Thought (CoT) or reflection mechanisms to explicitly "reason based on history and the map," the agents often diverge, reverting to a reactive, myopic reasoning pattern that ignores the very context they were instructed to use. This disconnect helps explain why adding CoT and reflection did not consistently improve performance (Table 2); the models did not faithfully execute the intended reasoning strategy. This suggests a significant gap between simply conditioning a model on a prompt and instilling a robust, procedural reasoning capability. Full CoT examples can be found in the supplementary materials.

**Spatial and Environmental Understanding.** Our fine-grained error analysis reveals that profound weaknesses in spatial understanding are the root cause of most navigational failures. Of 131 errors analyzed, a staggering 106 were due to persistent looping, a direct consequence of the model's inability to ground instructions in the 3D environment. This manifests in specific, recurring issues like poor region recognition (37 cases), failure to reason about verticality on stairs (30 cases), and basic directional confusion (11 cases). The fact that providing an explicit topological map failed to yield significant gains highlights a deeper problem: the agent cannot connect abstract spatial knowledge to its visual perception and actions. Furthermore, the agent critically fails at sequential decision-making, which is essential for navigation. The rampant looping behavior clearly shows that the agent does not learn from its trajectory to avoid repeating mistakes. This is not a problem of memory capacity, as the history rarely exceeds the model's context window, but rather one of memory utilization. The model has access to its past actions but cannot ground its current decisions in that history to self-correct. In fact, the observation that simpler history formats can outperform complex ones suggests that too much historical information creates a cognitive load, confusing the agent instead of guiding it.

**Perception-Action Grounding.** Finally, we observe a critical gap between multimodal perception and embodied action. The MLLM's visual grounding is functional at a recognition level; for example, it can often correctly identify a "staircase" or a target "chair" in its textual reasoning trace. This indicates that the visual and language modalities are connected. However, this recognition consistently fails to translate into correct action. The agent sees the stairs but walks past them in a loop. It may even get very close to the goal, demonstrating it has successfully grounded the target object visually, yet fails to execute the final 'STOP' action. This is powerfully illustrated by our success-case analysis, where 42 of 52 successful episodes involved inefficient looping near the target before stopping. This "perception-action gap" shows that the greatest challenge for MLLMs in VLN is not just seeing and describing the world, but effectively acting within it.

## 5 CONCLUSION

In this work, we investigate the performance of Multimodal Large Language Models (MLLMs) as zero-shot agents in Vision-and-Language Navigation (VLN). We introduce VLN-MME, a unified, modular, and simulator-free framework designed to systematically evaluate diverse MLLMs and agent architectures. Our analysis shows that current MLLMs are hindered by fundamental limitations in spatial reasoning and in translating perception into action, resulting in poor zero-shot performance. By enabling fine-grained error analysis, VLN-MME moves beyond simple success metrics to diagnose why agents fail, laying the groundwork for developing more capable embodied agents. We believe our analysis clearly reveals the error pattern for MLLM as a zero-shot navigation agent, and provides strong guidance for CoT reasoning data generation in VLM post-training as navigation agents.

## REPRODUCIBILITY STATEMENT

We are committed to ensuring the reproducibility of our research. Our primary contribution is the VLN-MME framework, a modular and simulator-free software stack designed specifically to facilitate standardized and reproducible evaluation of MLLMs in VLN tasks. To this end, we will make the following resources publicly available upon publication:

- **Source Code**: The complete source code for our evaluation framework, including implementations for all agent architectures, model interfaces, and evaluation scripts, will be released under a permissive open-source license.

- **Data and Environment**: All curated data splits from the R2R, REVERIE, and ObjectNav datasets used in our benchmark will be provided. This includes the pre-rendered panoramic observations, viewpoint connectivity graphs, and generated textual annotations (scene descriptions and captions) that enable our simulator-free approach.

- **Experimental Configurations**: The YAML configuration files for all experiments reported in this paper will be included, allowing for the exact replication of our results.

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
