# SUPPLEMENTARY MATERIAL FOR VLN-MME

Section A details the process for generating semantic annotations for the environment. The method for constructing agent-centric visual observations is described in Section B. Section C explains the complete prompt structure used for all agent variants. Our custom tool for trajectory visualization and analysis is introduced in Section D. Section E provides a detailed quantitative analysis of agent failures and successes. Finally, Section F illustrates common agent behaviors through several qualitative case studies. Section G shows the examples of CoT reasoning of MLLMs. Additionally, Section H includes our declaration on the use of large language models to aid in polishing the manuscript.

## A    GENERATION OF SEMANTIC ANNOTATIONS

To enrich the agent's environmental understanding in our simulator-free setup, we generated two types of semantic annotations: descriptive captions for navigable markers and concise summaries for each viewpoint.

### A.1    MARKER CAPTION GENERATION

The visual markers indicating navigable viewpoints in the panoramic images were annotated with short, descriptive captions. This process provides the agent with crucial semantic cues about the direction of potential paths. We used GPT-4o for this task. For each viewpoint, the model was provided with the panoramic image containing numbered visual markers and prompted to generate a JSON object mapping each marker index to a descriptive sentence. The prompt used for this captioning process was as follows:

> Observe the panoramic images provided, each labeled with distinct markers of green circles (indexed in number). For each marker, briefly specify the area or room it leads toward and describe what visually distinguishes it from the others in a short sentence. Present your response in JSON format, where each marker's index is a key and the corresponding short, descriptive sentence is the value.
>
> For example, if the image contains three markers, the response should look like:
> {
>     "1": "Decorative partition and dining area; leads toward an interior space or adjacent room.",
>     "2": "Chair and decorative cabinet area; leads to a wall-mounted decoration and seating.",
>     "3": "Hallway with external view and door; leads towards the entrance or exterior patio."
> }

Figure 1: GPT4o prompt for generating marker caption

### A.2    VIEWPOINT SUMMARY GENERATION

In addition to marker captions, a single, holistic summary of the scene was generated for each viewpoint to give map-based agents a global understanding of their current location. For this process, we adopt the same methodology presented in NavGPT (Zhou et al., 2024).

The generation follows a two-stage process. First, initial descriptions are generated for images from a viewpoint using the BLIP-2 model. To elicit descriptions that are rich in object details and relevant to indoor scenes, each image is fed to BLIP-2 with the prompt: "*This is a scene of*".

As this initial step often produces redundant information across different images of the same viewpoint, a second summarization step is employed. The descriptions generated by BLIP-2 (Li et al., 2023) are consolidated into a single, coherent sentence using a GPT-3.5 summarizer. The model is prompted with the following template in Figure 2, where "{description}" is replaced by the text from BLIP-2:

This two-stage approach ensures the final viewpoint summary is both informative and compact, ideal for inclusion in the agent's prompt history.

Here is a single scene view from top, down and middle: {description}
Summarize the scene in one sentence:

Figure 2: GPT4o prompt for generating viewpoint summarization

## B   CONSTRUCTION OF AGENT-CENTRIC VISUAL OBSERVATIONS

To provide the MLLM with a full 360-degree visual context from the agent's perspective, we construct a single panoramic image at each step. This process leverages the four pre-rendered, world-oriented images associated with each viewpoint and reorients them based on the agent's current heading. This method serves as a lightweight, simulator-free proxy for real-time rendering.

**Pre-rendered Image Set**   As described in the main paper, each viewpoint in the environment is associated with four high-resolution images, each with a 90-degree vertical Field of View (vFOV). These images are centered on the four cardinal directions relative to the global coordinate system: 0° (North), 90° (East), 180° (South), and 270° (West).

**Heading Correction and Image Selection**   Since an agent's heading is continuous (e.g., 60°), it will not always align perfectly with one of the four pre-rendered directions. To resolve this, we implement a heading correction mechanism. The agent's current continuous heading is first mapped to the closest cardinal direction. This is achieved by quantizing the heading angle to the center of the 90-degree quadrant it falls within. For instance, any agent heading $h \in [45, 135)$ is mapped to the 90° image, which then serves as the agent's **Front** view.

**Panoramic Image Composition**   Once the **Front** image is determined through heading correction, the remaining three images are assigned to the agent's relative directions: **Left**, **Right**, and **Back**. These four images are then concatenated horizontally in the following order to form a single panoramic strip: [**Left**, **Front**, **Right**, **Back**].

To ensure the MLLM can correctly interpret this composite view, we explicitly annotate the image by overlaying the corresponding directional labels above each of the four segments, as illustrated in Figure 3. This provides a clear, agent-centric visual input that grounds the model in its current orientation.

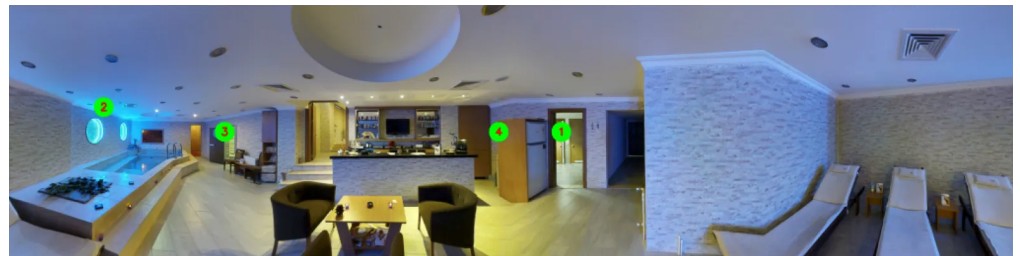

Figure 3: An example of the composite visual observation provided to the MLLM. The four pre-rendered images are stitched together in an agent-centric order (Left, Front, Right, Back) based on the agent's corrected heading.

## C   AGENT PROMPT DESIGN

A central component of our framework lies in the design of prompts that guide multimodal large language models (MLLMs) to behave as navigation agents. Since the main paper provides only

a brief overview, we expand here with a full account of the structure, components, and variations used across all eight agent types. Our agents are divided into two families: *Text Summarization as Memory (NavGPT)*, which relies purely on local observations and history, and *Text Map as Memory (MapGPT)*, which augments navigation with dynamically constructed topological maps. Within each family, we instantiate four variants: a baseline version, a chain-of-thought (CoT) agent, a reflection-enabled agent, and a combined CoT+Reflection agent. This section explains the design philosophy of each family and the detailed structure of their prompts.

## C.1 PROMPT STRUCTURE

All prompts are composed of two distinct parts: the *system* and the *task* component. The system portion defines the global context of the agent, introducing the VLN setting, enumerating the input elements, and stating the rules the model must follow when reasoning about navigation. It also enforces the strict output format required for downstream evaluation. The task portion is dynamic, providing the specific input to the agent at each time step: the instruction, navigation history, agent orientation, and the set of navigable options. Together, these two components establish both the constraints and the situational awareness necessary for coherent decision making. Figure 4 illustrates an example of the full text summarization as memory baseline prompt.

```
[System Prompt]
You are a Vision-Language Navigation (VLN) agent in an indoor environment.
Your task is to select ONE next action that follows the navigation instruction, based on:
1. Observation - A panoramic image at the current location (4 views: left, front, right,
back) with numbered green-circle markers for possible moves.
2. Instruction - Step-by-step route. Some steps may already be completed; focus only
on the remaining.
3. History - Past moves, including mistakes and loops.
4. Action Options - A dictionary of possible movement choices grouped by direction.
Each option has a unique number as its ID and a text description. The direction names
(Left, Front, Right, Back) are for grouping only — they are NOT part of the output.

Key Rules:
- Avoid loops: Do not repeat recent viewpoints or oscillate unless required.
- Do not revisit: If a viewpoint was visited multiple times, stop instead.
- Detect arrival: If the scene matches the destination in the instruction, choose 'Stop'.
- Prefer progress: Select new paths that clearly advance toward the goal.
- Follow heading: Use current heading/elevation when interpreting the instruction.

Output format:
Action: <Option_ID>. <Option_description>
  - <Option_ID> must be ONLY the numeric ID from the action options (e.g., '3', '4').
  - DO NOT include the direction name in the output.
If stopping:
Action: Stop. Have reached the destination and stop here.

[Navigation Prompt]
Instruction: {instruction}
History: {history}
Current heading: {heading}°, elevation: {elevation}°
Action options: {action_options}

Choose the best next action ID and its description.
Return the result in the exact format specified above.
```

Figure 4: Text summarization memory baseline agent prompt structure

## C.2 Text Summarization as Memory Agents

The NavGPT-style agents are designed to operate with information that would be available in a simulator-based VLN setup but translated into our simulator-free representation. The system prompt explicitly instructs the model to select a single next action, referencing only the option identifiers, and to obey a series of rules that reduce common navigation errors such as looping, oscillation, and premature stopping.

The task inputs are carefully structured. The navigation **instruction** is passed in verbatim, ensuring the model has access to the original language guidance. The **history** describes prior movements in natural language, with each step recorded as a turning angle, forward displacement, and the semantic description of the destination viewpoint. This representation provides both spatial reasoning cues and semantic grounding. The agent's **current heading and elevation** are provided as numerical values, anchoring the model's interpretation of orientation. Finally, the **action options** are represented as a dictionary keyed by relative directions, where each entry contains a marker ID and a semantic description of the corresponding navigable viewpoint, along with an explicit "Stop" action. This structured but naturalistic representation ensures the model can ground its decisions in both geometry and semantics.

## C.3 Reasoning-Enhanced Text Summarization Memory Agent Variants

To probe the role of explicit reasoning, we introduce three reasoning-augmented variants of text summarization memory agent. In the CoT version, the system prompt is modified so that the agent first produces a reasoning trace encapsulated in <Reasoning> tags before committing to its final action choice. This design encourages more transparent step-by-step deliberation. The Reflection variant modifies the output format further: after producing an action, the agent generates a reflective evaluation wrapped in <Reflection> tags, followed by a <Final Decision> statement declaring whether to keep or revise its action. If the reflection deems the decision unsound, the agent replans rather than moving. The CoT+Reflection version combines both mechanisms, first reasoning explicitly and then reflecting on the proposed choice, providing the richest form of introspective navigation. These modifications shift the model from direct action prediction toward a more deliberative, self-monitoring behavior.

## C.4 Text Map as Memory Agents

While NavGPT-style focuses on local decision making, the MapGPT-style agents introduces a form of spatial memory through a dynamically constructed topological graph. At each step, the agent augments its prompt with a **map connectivity** field, expressed in natural language, that lists adjacency relationships between viewpoints (e.g., "node_0 is connected to node_1, node_2"). This evolving graph representation enables the MLLM to reason not only about immediate action choices but also about the broader connectivity of the explored environment.

The navigation history for text map memory agents is likewise enriched. Instead of recording only motion trajectories, it includes the current node identifier, a semantic description of the viewpoint, and the sets of visited and unvisited nodes. This structure gives the agent both a local semantic grounding and a global perspective on the exploration state. A complete prompt example after one navigation step is illustrated in Figure 5.

## C.5 Reasoning-Enhanced Text Map Memory Agent Variants

The CoT, Reflection, and CoT+Reflection augmentations are applied to text map memory agents in the same manner as for text summarization agens, modifying only the output structure while retaining the additional map input. Thus, the text map memory agents explores how explicit reasoning interacts not just with semantic cues, but also with global topological memory.

# D Trajectory Visualization and Analysis Tool

This section details the custom tool developed for the qualitative analysis of agent trajectories. The tool, named the VLN Result Visualizer, developing using Gradio, provides an interactive interface

**Instruction:** Walk down the stairs. Walk forward and stop next to the door that is next to the recycling bin.

**History:**
Navigation starts.
Step 1: Turning heading direction -37.78 degrees from right 36.90 to left 0.88, and forward 3.42 meters towards Staircase with glass panels and wall-mounted artwork; leads toward an upper interior area or room.

**Current heading:** -0.88°, **elevation:** 27.21°

**Action options:**
```
{
  "Left": {
    "3": "Glass-enclosed gym area with exercise equipment; leads toward the fitness room.",
    "4": "Open gym area with visible workout machines; leads toward the main exercise space.",
    "5": "Brightly lit gym area with additional equipment; leads further into the fitness zone."
  },
  "Front": {
    "1": "Hallway with a brown wall and utility area; leads toward a corridor or storage space."
  },
  "Right": {},
  "Back": {
    "2": "Wall with artwork and a staircase; leads toward a lower level or exit."
  },
  "Stop": "Have reach the destination and stop here."
}
```

Choose the best next action ID and its description.
Return the result in the exact format specified above.

Figure 5: Text map memory agent prompt example (Step 1)

for a step-by-step inspection of any navigation episode, which is crucial for understanding the nuances of agent behavior beyond aggregate metrics.

The visualizer is built entirely using the Gradio framework. Its primary function is to parse the evaluation result files and present the information in a human-readable format. At the top of the interface, a user can specify the configuration used during evaluation, including the **agent type**, **MLLM model**, **task**, and **data split**. Once a configuration is loaded, a dropdown menu is populated with all episode IDs from that run, allowing for the selection of any specific trajectory for analysis.

The core of the interface, shown in Figure 6, is the visual observation panel. It displays the agent's panoramic view for the **Current Viewpoint** and, if a valid move is made, the panoramic view of the chosen **Next Viewpoint**. Each panoramic image is a composite of four individual images presented in the agent-centric order of [**Left**, **Front**, **Right**, **Back**], with the global orientation (e.g., "View: Right") explicitly labeled above each segment. Navigable options are clearly marked with green circular markers.

Below the visual panel, detailed textual information for the current step is provided. This includes: the **step number**, the original **navigation instruction**, the raw **LLM Output**, the parsed **agent action** (turn angle and forward distance), the **full trajectory** path taken so far, and the **ground truth path**. Critically, the tool also flags the exact step at which the agent's path first **deviated from the ground truth**, enabling quick identification of crucial mistakes. This is followed by the complete **history log** that was fed into the model at that step, allowing for an in-depth analysis of the agent's reasoning context.

To further enhance usability, the interface provides status indicators directly on the display during navigation. The ID of the **current viewpoint** and the numeric ID of the **next chosen marker** are displayed at the top of the screen, providing immediate context without needing to consult the text logs.

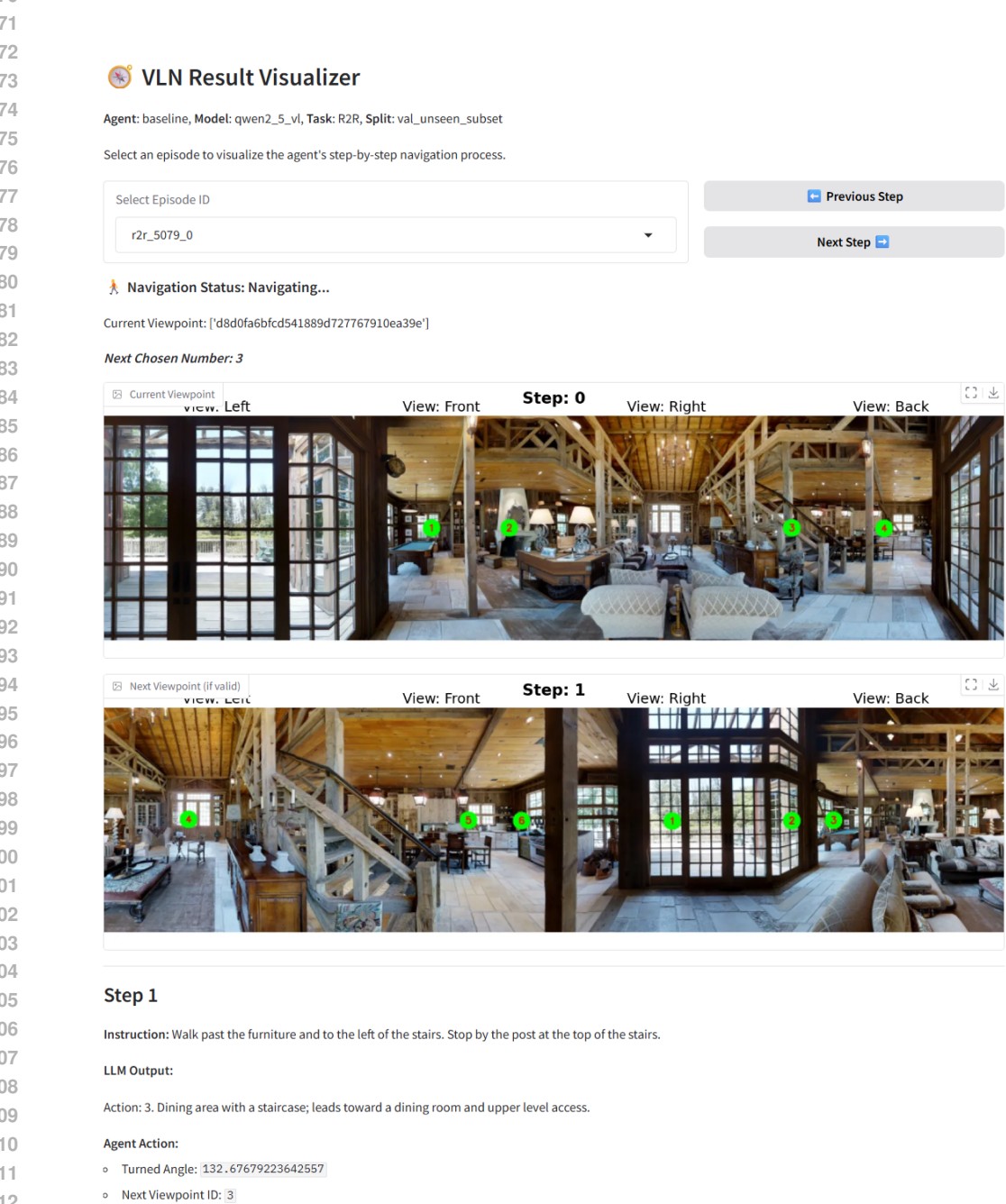

Figure 6: The main interface of the VLN Result Visualizer.

At the conclusion of each episode, a final summary panel presents the quantitative **evaluation metrics**, such as Success, SPL, Navigation Error, and Trajectory Length, offering a direct link between the agent's step-by-step actions and its final performance score. This tool was indispensable for conducting the detailed error analysis presented in this paper.

## E    DETAILED ERROR AND SUCCESS ANALYSIS

This section provides a detailed analysis of agent performance across 200 navigation trajectories. By breaking down both failures and successes, we can identify the primary challenges MLLM-based agents face in the VLN task.

The results reveal a significant performance gap, with 148 failures compared to only 52 successes. An initial breakdown of the failures, as shown in Figure 7, indicates that the vast majority (131 out of 148) stem from **incorrect navigation** rather than technical **MLLM Generation Errors** (17 cases). This suggests that while the models are generally capable of producing valid actions, their decision-making logic is the primary point of failure.

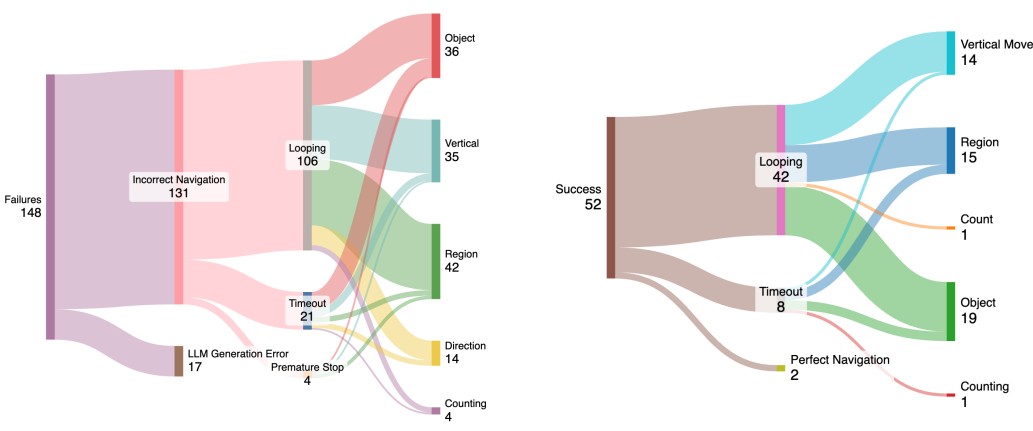

Figure 7: Analysis of error sources in 148 failure episodes.

Figure 8: Analysis of navigation behavior in 52 successful episodes.

Within the incorrect navigation errors, **looping** is the most dominant failure mode, accounting for a remarkable 106 cases. This behavior, where the agent repeatedly revisits the same viewpoints, points to a fundamental difficulty in spatial awareness and state tracking. The root causes for these loops, as well as for timeouts, are primarily failures in high-level scene understanding. Specifically, **region recognition** (37 cases in looping), **vertical movement understanding** (30 cases), and **object detection** (25 cases) are the most frequent triggers for getting stuck. This highlights the agent's struggle to match abstract instructions (e.g., "go to the kitchen", "go upstairs") with visual evidence.

Conversely, an analysis of the 52 successful trajectories provides a more nuanced picture of the agent's capabilities, as illustrated in Figure 8. A striking finding is that only **2 trajectories were completed perfectly**. The vast majority of successes (42 cases) were achieved despite the agent exhibiting **looping behavior**, typically near the target. This suggests that while agents can eventually recover from local confusion, their navigation is often highly inefficient. The challenges in these near-success cases mirror those in the failures: difficulties with **object recognition** (16 cases), **vertical movement** (13 cases), and **region understanding** (11 cases) still persist, causing inefficiency even when the final goal is reached.

In conclusion, the data indicates that the primary obstacle for these MLLM agents is not language generation but robust **spatial and semantic reasoning**. The pervasive issue of looping, both in failed and successful episodes, underscores a weakness in creating and maintaining a stable understanding

of the environment. Future work should focus on enhancing these core reasoning capabilities to improve both the reliability and efficiency of navigation.

# F  CASE STUDIES

In this section, we provide a qualitative analysis of five navigation episodes to illustrate the agent's common behaviors, highlighting both its capabilities and frequent failure modes.

Figures 10 and 11 showcase successful episodes that also reveal subtle inefficiencies. In Figure 10, the agent correctly identifies the target treadmill but exhibits redundant behavior by moving away and looping back before executing the final stop action. Similarly, the episode in Figure 11 demonstrates a strong recovery capability, yet the agent struggles with precise vertical positioning, causing it to loop on the stairs rather than stopping at the correct step.

Conversely, Figures 12, 13, and 14 depict common failure scenarios. The trajectory in Figure 12 represents a case of 'oracle success,' where the agent navigates to the immediate vicinity of the destination downstairs but ultimately fails by getting trapped in a repetitive loop on the staircase. Figure 13 illustrates a multi-faceted failure; the agent not only fails to ground the directional instruction and identify the 'hallway' but also produces a malformed output, resulting in an un-parsable command and an invalid action error. Finally, Figure 14 demonstrates a failure in semantic region understanding, where the agent is unable to correctly interpret the goal of stopping 'inside of the sauna'.

# G  EXAMPLE CoT REASONING

```
<Reasoning>:
1. Progress estimate: The instruction is to walk up stairs and wait at the top.
The current observation shows a staircase leading upward, which aligns with the
instruction.
2. Further analysis: The current heading and elevation suggest we are at the
top of the stairs, as there are no more stairs visible ahead. The back view
confirms we are at the top, as the glass-walled corridor leads to an exit or
open area, not further stairs.
3. Prediction: Since the instruction specifies waiting at the top, and the
observation confirms we are at the top, the next logical action is to stop.
</Reasoning>

----------------------------------------------------------------

<Reasoning>:
1. Progress estimate: The instruction requires going up the stairs and turning
left at the top. The current observation shows the staircase leading upward,
which aligns with the first part of the instruction.
2. Further analysis: The current heading is 93.59°, which suggests we are
facing the staircase. The left option indicates a staircase with wall-mounted
artwork, which is consistent with the instruction to turn left at the top.
3. Prediction: The next logical step is to move towards the left staircase to
continue following the instruction.
</Reasoning>
```

Figure 9: Two examples of chain-of-thought (CoT) reasoning generated by Qwen2.5-VL-7B. Both cases demonstrate structured step-wise reasoning but limited integration of historical context.

Figure 9 presents two representative CoT outputs. In both cases, the model correctly decomposes the navigation instruction into progress estimation, further analysis, and prediction. However, despite being provided with full interaction history, the reasoning predominantly relies on local observations and the most recent instruction, while largely neglecting prior context.

This illustrates a broader issue: even when the input tokens are well within the context window, the model exhibits poor reasoning fidelity across multiple rounds of context. Instead of leveraging accumulated history for richer reasoning, the MLLM tends to perform single-turn grounding of the immediate observation. This behavior highlights a critical limitation for VLN tasks, where successful navigation often depends on integrating long-term history with dynamic, stepwise decision-making.

# H  The Use of Large Language Models (LLMs)

As disclosed, we utilized LLMs (GPT5, Google Gemini etc.) to aid in polishing the manuscript's prose. Its role was to improve grammatical correctness and sentence clarity, with all final content being reviewed and approved by the authors, who take full responsibility for this work.

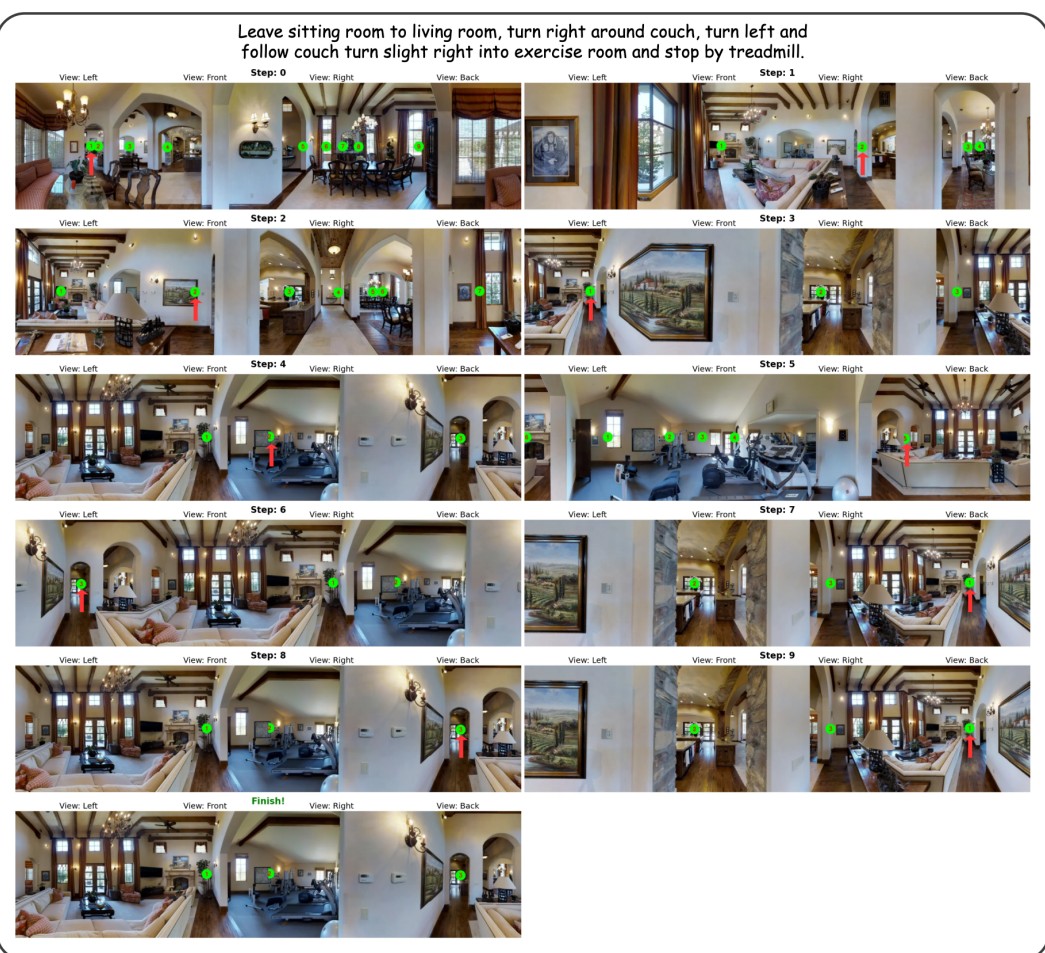

Figure 10: A successful but inefficient trajectory. After observing the target treadmill, the agent loops around the room before stopping.

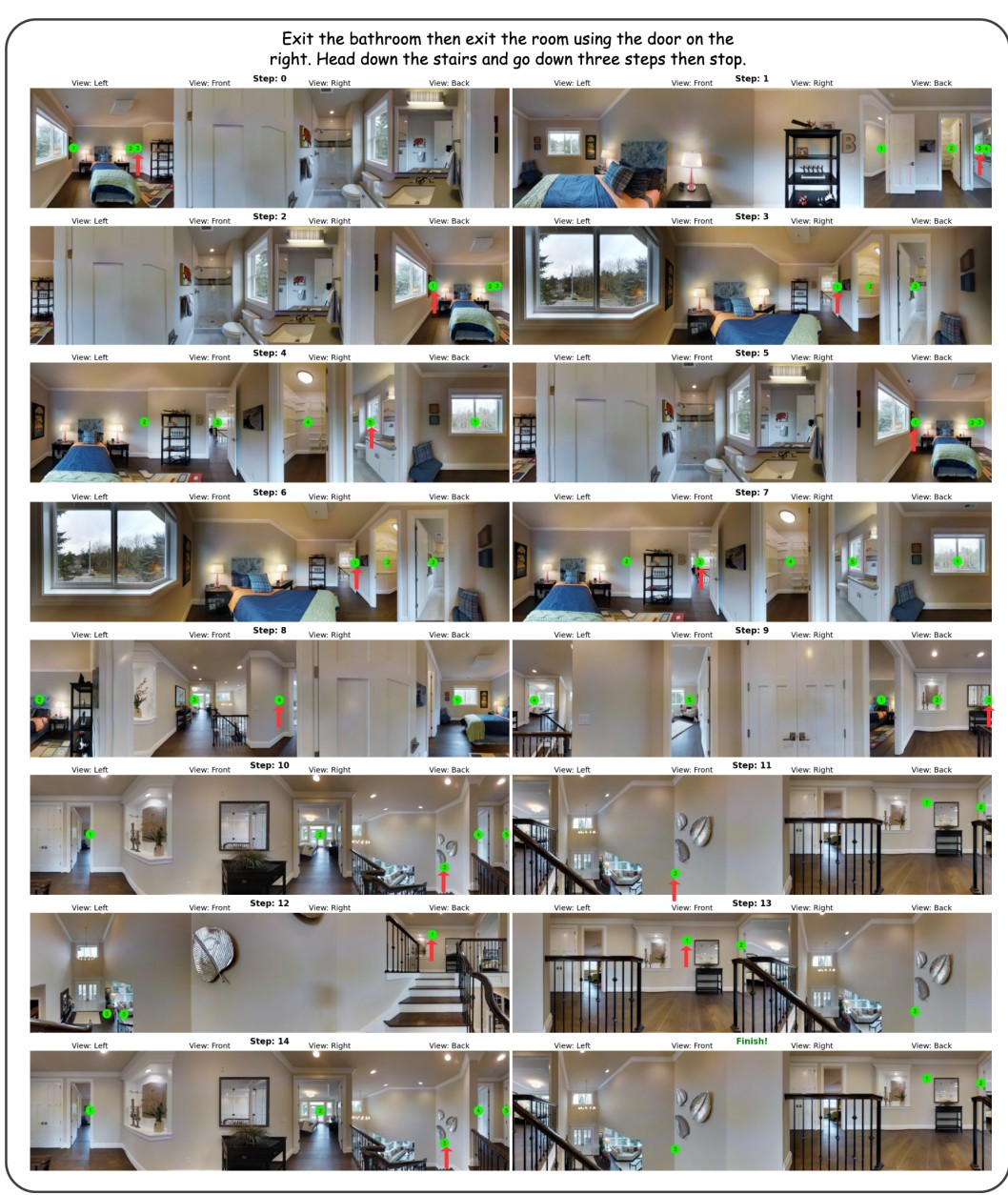

Figure 11: A successful episode showcasing recovery capabilities. However, the agent exhibits looping behavior during vertical movement on the stairs.

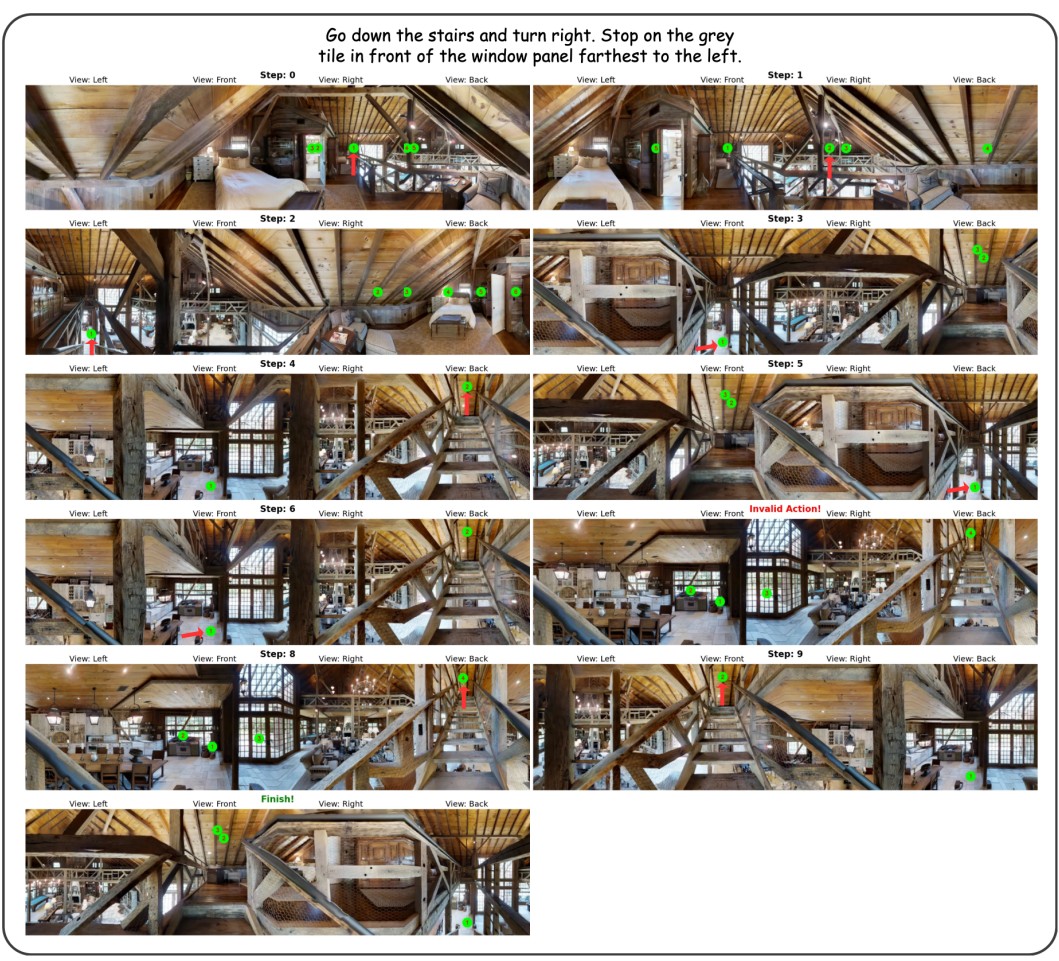

Figure 12: A failure case with oracle success. The agent reaches the correct general area but fails to stop, getting stuck in a loop on the stairs.

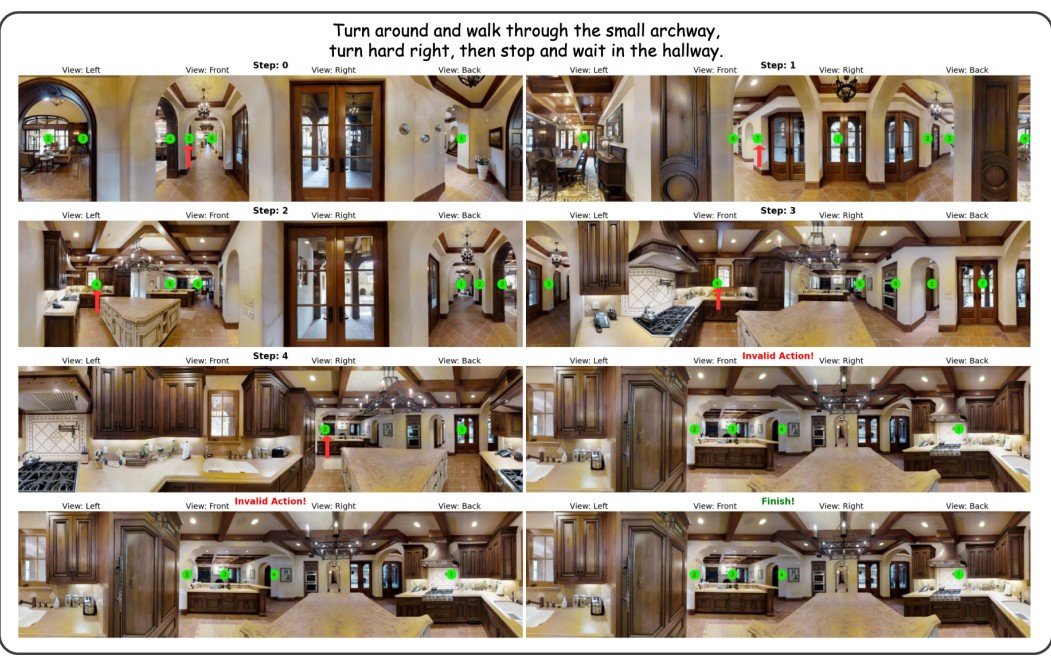

Figure 13: Navigation failure due to misinterpreting a directional instruction and a model generation error that produced an invalid action.

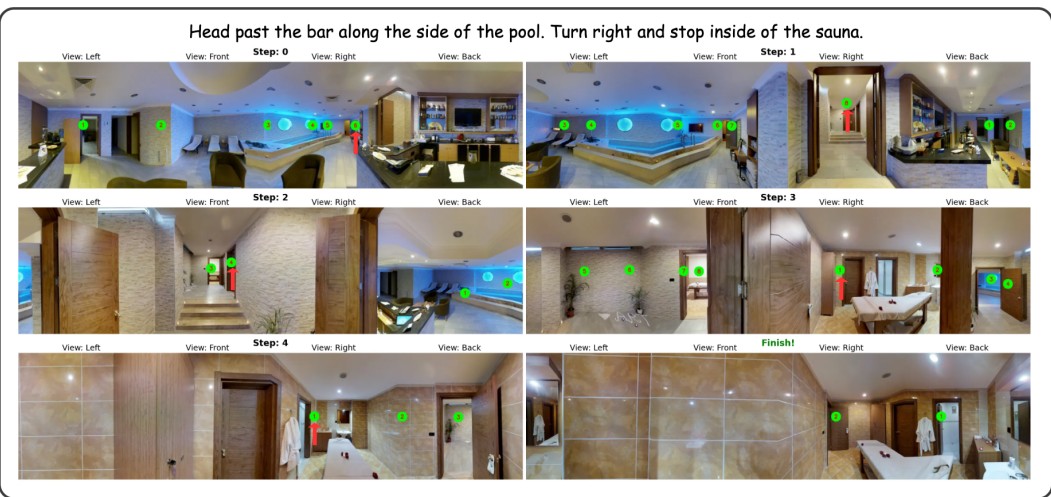

Figure 14: A failure episode caused by the agent's inability to understand and navigate into the specified target region ('sauna').