# OpenReview forum: "VLN-MME: Diagnosing MLLMs as Language-guided Visual Navigation agents"
_ICLR.cc/2026/Conference — ICLR 2026 Conference Withdrawn Submission_

### Official Review · Reviewer_MRS8 · 2025-10-27

**Soundness:** 4
**Presentation:** 4
**Contribution:** 3
**Rating:** 6
**Confidence:** 3

**Summary:**

The authors present VLN-MME, a benchmark framework for multimodal-LLMs on vision and language navigation (VLN) tasks. The framework provides an interface for tasks, agents and models, such that a new methods can be evaluated in a standardized setting. The tasks include REVERIE, ObjectNav and R2R, three commonly used VLN tasks. A welcomed addition are the prerendered images for each viewpoint in the environments, such that costly and task-specific environment rendering is abstracted away. The framework also provides a fixed subset of representative routes for each task such that the full suite can be run in a reasonable amount of time. The provides baseline agents and granular evaluation metrics will make it straight forward to benchmark new MLLMs.

**Strengths:**

- simulation/rendering free evaluation
- baseline agents
- fine-grained metrics

**Weaknesses:**

- limited set of tasks. Could include outdoor VLN [1,2] for an additional distinct setting.


[1] Touchdown: Natural Language Navigation and Spatial Reasoning in Visual Street Environments, Chen et al., 2018
[2] VELMA: Verbalization Embodiment of LLM Agents for Vision and Language Navigation in Street View, Schumann et al., 2024

**Questions:**

- Why GPT-4o is used to pre-generate textual descriptions of the scene instead of making this part of the evaluation/benchmark?
- Does prerendering limit this benchmark to discrete navigation instead of taking continuous actions?

---

> ### Author Response · Authors · 2025-11-24
> **Author Responses to Reviewer MRS8**
>
> **W1 - Limited set of tasks. Could include outdoor VLN [1,2] for an additional distinct setting:**
>
> We thank the reviewer for this suggestion. However, we must clarify that this perceived limitation is a deliberate and necessary experimental design choice.
>
> The explicit objective of our benchmark is to evaluate an MLLM's ability to understand fine-grained 3D environments and make sequential decisions. Our work is specifically positioned to analyze tasks traditionally studied within simulators like MP3D and Habitat.
>
> While outdoor VLN (as in Touchdown) is a valid task, it is not amenable to the precise, quantitative navigation metrics (e.g., Success Rate (SR), OSR, SPL, and nDTW) that are standard in indoor VLN. These metrics are essential to our work, as they allow us to robustly quantify performance and discover the specific failure modes of MLLM reasoning.
>
> Therefore, we intentionally limited our scope to indoor environments to maintain this high-resolution evaluation. Thus, this is a focused approach to isolate and rigorously measure the core reasoning abilities we claim to study.
>
> **Q1 - Why GPT-4o is used to pre-generate textual descriptions of the scene instead of making this part of the evaluation/benchmark?:**
>
> We thank the reviewer for this question, we could definitely include this as part of evaluation. But in this work, by using high-quality descriptions from a state-of-the-art model like GPT-4o, we provide a consistent, "gold-standard" goal instruction to all agent models. This ensures that any failures we observe are due to the agent's planning and reasoning limitations, not its (separate) ability to describe a scene. This isolation of the planning task is critical for the validity of our conclusions. Additionally, we do conduct experiments on SOTA models like GPT5, and for the detailed information, please refer to **General Response: Test result on SOTA models**.
>
> **Q2 - Does pre-rendering limit this benchmark to discrete navigation instead of taking continuous actions?:**
>
> We thank the reviewer for this question. Yes, our pre-rendering design does limit to discrete navigation setting. However, the main purpose of our benchmark is to evaluate an MLLM's ability to perform 3D environment understanding, spatial reasoning, historical understanding, and sequential decision-making. A discrete action space is entirely sufficient to rigorously test these core cognitive abilities. For the more detailed information, please refer to **General Response: Contribution**.

---

### Official Review · Reviewer_MoHw · 2025-10-29

**Soundness:** 3
**Presentation:** 2
**Contribution:** 2
**Rating:** 2
**Confidence:** 3

**Summary:**

This paper introduces a modular, simulator-free framework for evaluating MLLMs as zero-shot agents in vision-and-language navigation. It standardizes agents, models, and tasks; supports representative R2R/REVERIE/ObjectNav datasets; and leverages pre-rendered, annotated panoramas for fine-grained diagnosis. Experiments show that while zero-shot MLLMs set useful baselines, they trail finetuned specialists, and Chain-of-Thought often fails to improve performance.

**Strengths:**

* Studying MLLMs as embodied agents for language-guided visual navigation is an important research direction.

* The proposed framework provides a practical and effective way to evaluate MLLM-based agents.

* Its simulator-free design significantly reduces the computational cost of simulation.

**Weaknesses:**

* Overall, the paper’s contribution is quite limited. Evaluation of MLLM-based navigation agents already appears in EmbodiedBench [1] and EmbodiedEval [2], which cover diverse scenes, instructions, and difficulty levels, and already support modular MLLM evaluation in EmbodiedBench. The paper does not clearly differentiate itself from these closely related efforts, so the true contribution is unclear and quite limited to my understanding.

* The paper uses three datasets, R2R, REVERIE, and ObjectNav, but does not explain their differences or the rationale for selecting them. A brief, focused introduction and justification would help.

* While the simulator-free setup reduces cost, it constrains agents to predefined actions that may not reflect real-world behavior. This raises concerns about realism and the size of the sim-to-real evaluation gap.

* In addition, I think this paper does not provide enough insight for agent design as claimed. Beyond noting that CoT/Reflection often underperform, the paper does not analyze why or propose concrete ways to improve them. More diagnostic evidence and actionable guidance are needed from the large table.

[1] EmbodiedBench: Comprehensive Benchmarking Multi-modal Large Language Models for Vision-Driven Embodied Agents, 2025.

[2] EmbodiedEval: Evaluate Multimodal LLMs as Embodied Agents, 2025.

**Questions:**

Please refer to the Weaknesses part.

---

> ### Author Response · Authors · 2025-11-24
> **Author Responses to Reviewer MoHw**
>
> **W1 - Overall, the paper’s contribution is quite limited. Evaluation of MLLM-based navigation agents already appears in EmbodiedBench [1] and EmbodiedEval [2], which cover diverse scenes, instructions, and difficulty levels, and already support modular MLLM evaluation in EmbodiedBench. The paper does not clearly differentiate itself from these closely related efforts, so the true contribution is unclear and quite limited to my understanding:**
>
> We thank the reviewer for this opportunity to clarify our contribution relative to *EmbodiedBench* and *EmbodiedEval*. While we agree these are valuable broad-spectrum benchmarks, our work serves a fundamentally different purpose. Unlike their generalist approach using synthetic simulators, our framework is a **specialized, simulator-free diagnostic tool** built on realistic 3D scans. Our contribution lies in providing a lightweight, reproducible setting to strictly isolate and diagnose MLLM spatial reasoning failures, rather than evaluating general embodied tasks. We have provided a detailed comparison regarding scope, environment, and goals in the **General Response: Contribution, Compare to embodied benchmarks**.
>
> **W2 - The paper uses three datasets, R2R, REVERIE, and ObjectNav, but does not explain their differences or the rationale for selecting them. A brief, focused introduction and justification would help:**
>
> We thank the reviewer for this suggestion. This was a deliberate choice to ensure our benchmark evaluates MLLM capabilities across a comprehensive spectrum of instructional granularities and task objectives.
>
> The rationale for selecting these three specific datasets is as follows:
>
> - R2R (Room-to-Room): Provides fine-grained, step-by-step instructions. This tests the agent's ability to follow a detailed, prescribed path and execute low-level commands.
> - REVERIE (Remote Embodied Visual Referring Expressions): Provides coarse-grained, high-level instructions. The agent is told *where* to go (e.g., "go to the kitchen") but not *how* to get there. This tests the agent's ability to perform autonomous, high-level planning and reasoning to fill in the missing steps.
> - ObjectNav (Object Goal Navigation): Provides no path instructions and description of object at all (zero-grained, e.g., "find the chair"). This represents the most ambiguous case, testing the agent's ability to perform autonomous exploration, scene understanding, and search based only on a target category.
>
> By evaluating across these three distinct paradigms, from fully-guided to coarse-guided to zero-guidance, our benchmark can provide a much richer, more nuanced analysis of an MLLM's true planning and reasoning capabilities. We will clarify this justification in the paper.
>
> **W3 - While the simulator-free setup reduces cost, it constrains agents to predefined actions that may not reflect real-world behavior. This raises concerns about realism and the size of the sim-to-real evaluation gap:**
>
> We thank the reviewer for this comment. We respectfully clarify that our use of predefined actions **aligns with the standard discrete VLN setting** widely adopted in the field (e.g., R2R, RxR, REVERIE). This design choice ensures our benchmark remains consistent with established problem definitions, allowing for fair comparison of high-level planning logic without the confounders of low-level continuous control. Our objective is to strictly isolate and evaluate **cognitive abilities,** specifically 3D spatial reasoning and sequential decision-making, rather than sim-to-real physics transfer. For a further discussion on these design choices, please refer to the **General Response: Contribution**.
>
> **W4 - In addition, I think this paper does not provide enough insight for agent design as claimed. Beyond noting that CoT/Reflection often underperform, the paper does not analyze why or propose concrete ways to improve them. More diagnostic evidence and actionable guidance are needed from the large table:**
>
> We thank the reviewer for this comment. We respectfully clarify that our **benchmark and diagnostic analysis could definitely** guide agent design.
>
> We respectfully remind the reviewer that in our error analysis, we observed that MLLMs struggle with specific spatial concepts, such as vertical transitions (e.g., going up stairs) and region recognition (Sec 4.3). Motivated by this insight, we conducted additional experiments using **Oracle Assistance** and **Failure-Aware Few-Shot prompting**. These experiments prove that the benchmark can successfully isolate latent MLLM capabilities and identify reasoning bottlenecks. For detailed results, please refer to the **General Response: Oracle Assist and Few-Shot Experiments**.

---

### Official Review · Reviewer_HHTS · 2025-11-01

**Soundness:** 3
**Presentation:** 2
**Contribution:** 2
**Rating:** 2
**Confidence:** 4

**Summary:**

The paper introduces VLN-MME, a modular and simulator-free benchmark and evaluation framework to assess MLLM as zero-shot navigational agents in VLN tasks. The framework emphasizes diagnostic analysis beyond aggregate metrics, supports diverse agent/model architectures, and provides a lightweight evaluation through pre-rendered panoramic data. Experimental analysis compares several open-source MLLMs and agent prompting strategies, revealing unexpected performance degradations when incorporating COT reasoning and self-reflection. Detailed error analyses highlight fundamental gaps in spatial reasoning and perception-action grounding for these models.

**Strengths:**

1. The paper proposes a well-conceived and extensible evaluation suite for VLN tasks, which is a practical tool for the community. The modularity is clearly described, and the simulator-free design reduces computational burden, lowering the barrier to entry for benchmarking and reproducibility.
2. The paper is well structured and easy to follow. The benchmark design, experiments, and analysis are presented in a clear and logical way.

**Weaknesses:**

1. The method section focuses on modular and simulator-free design. While I acknowledge the design efforts, I feel they are more like engineering work rather than scientifically driven research. VLN tasks have been deeply studied. I appreciate that the authors probably wrapped the evaluations into easy-to-use APIs, but I feel the this work is not substantially different in identifying the key capabilities of VLMs compared to other existing VLN benchmarks.

2. The authors use four pre-rendered, non-overlapping perspective images at each location to achieve a simulator-free setup. However, this design essentially functions as a simulator or a “space-for-time” trick. Moreover, it likely results in a fixed step length for the agent, making the setting less aligned with real-world conditions.

3. There is a noticeable absence of rigorous algorithmic or theoretical detail. Critical aspects such as prompt composition (order, length limits, input formatting), failure-handling policy specifications, and sampling protocols for negative/ambiguous visual cues in navigation are not formalized mathematically.

4. The paper does not evaluate proprietary models such as GPT-5, and most of the open-source models tested are under 10B parameters. I wonder whether simply scaling up the model size or using more powerful models could solve this benchmark.

**Questions:**

Please see weakness section.

---

> ### Author Response · Authors · 2025-11-24
> **Author Responses to Reviewer HHTS**
>
> **W1 - The method section focuses on modular and simulator-free design. While I acknowledge the design efforts, I feel they are more like engineering work rather than scientifically driven research. VLN tasks have been deeply studied. I appreciate that the authors probably wrapped the evaluations into easy-to-use APIs, but I feel the this work is not substantially different in identifying the key capabilities of VLMs compared to other existing VLN benchmarks:**
>
> We thank the reviewer for this comment. We respectfully argue that our intention is not to propose a VLN Benchmark.
>
> Previous benchmarks heavily rely on online rendering, are complex and time-consuming. This overhead creates a significant bottleneck that makes the rapid, reproducible, and large-scale evaluation of MLLMs impractical. Our framework enables the community, to finally focus on the scientific question that was previously too difficult to isolate.
>
> For detailed information please refer to **General Response: Contribution, Compare to existing MLLM benchmarks, Compare to embodied benchmarks.**
>
> **W2 - The authors use four pre-rendered, non-overlapping perspective images at each location to achieve a simulator-free setup. However, this design essentially functions as a simulator or a “space-for-time” trick. Moreover, it likely results in a fixed step length for the agent, making the setting less aligned with real-world conditions:**
>
> We would kindly correct the reviewer that our benchmark exactly follows the standard VLN setup, and the agent could dynamically interact with the environment, not “**results in a fixed step length**”. By saying simulation-free, we mean to skip the time for rendering image observations.
>
> **W3 - There is a noticeable absence of rigorous algorithmic or theoretical detail. Critical aspects such as prompt composition (order, length limits, input formatting), failure-handling policy specifications, and sampling protocols for negative/ambiguous visual cues in navigation are not formalized mathematically:**
>
> We thank the reviewer for this request for clarification. We provide details on these points below.
>
> 1. On Prompt Composition: We respectfully point the reviewer to Appendix B of our supplementary material, where we provide complete, verbatim details of all prompt compositions, including formatting, ordering, and lengths.
>
> 2. On Failure-Handling Policy: We have indeed implemented a rigorous policy for parsing the MLLM's output. Our system employs a failure-aware parsing mechanism that treats the MLLM's text output as a noisy, imperfect instruction, not a perfect command. It uses robust, pattern-agnostic extraction to recover the agent's *intent* (e.g., "go to viewpoint 3") from imperfect text, rather than failing on minor formatting errors. When parsing does fail, the system does not crash; it surfaces the failure as a structured signal ("Parse_Failed"). This transforms parsing errors from system crashes into measurable diagnostic events, which is critical for robust evaluation.
>
> 3. On Sampling Protocols: We respectfully point the reviewer to Section 3.2 (Lines 232-243), where our sampling protocol is detailed. Following previous work, we created a representative subset for evaluation.
>
> Critically, this was not a random sample. We employed a stratified sampling approach to ensure our subset's distributions for key characteristics (e.g., path length, instruction difficulty) mirror those of the full, original dataset.
>
> While a rigid mathematical formalization to sample for subjective, semantic properties like "ambiguity" or "negative cues" is often impractical (and not a standard for the original R2R/REVERIE benchmarks themselves), our stratification method ensures that we naturally capture a representative sample of these challenging cases inherent to the full dataset. This is a far more rigorous and practical approach than simple random sampling.
>
> **W4 - The paper does not evaluate proprietary models such as GPT-5, and most of the open-source models tested are under 10B parameters. I wonder whether simply scaling up the model size or using more powerful models could solve this benchmark:**
>
> We appreciate the reviewer for bringing this point up. For the detailed information, please refer to **General Response: Test result on SOTA models**.

---

### Official Review · Reviewer_B8ug · 2025-11-01

**Soundness:** 3
**Presentation:** 3
**Contribution:** 2
**Rating:** 4
**Confidence:** 4

**Summary:**

This paper investigates the capabilities of Multimodal Large Language Models (MLLMs) as zero-shot agents in Vision-and-Language Navigation (VLN). The authors introduce VLN-MME, an evaluation framework designed to be modular, extensible, and simulator-free to enable fast and accessible benchmarking. This framework is used to systematically evaluate various open-source MLLMs and agent architectures. The most significant and counter-intuitive finding is that applying advanced prompting strategies like Chain-of-Thought (CoT) and self-reflection consistently degrades performance. The authors attribute this to the models' poor context awareness and inability to ground reasoning in the sequential, embodied nature of the task.

**Strengths:**

1. **The paper is well-written.** The proposed method is well-illustrated and easy to follow.

2. **The simulator-free approach, which pre-renders observations and metadata, is evaluation-friendly.** It lowers the computational barrier to entry for VLN research.

3. **Insightful findings:** The central finding—that CoT and reflection-based reasoning harm performance—is counter-intuitive and impactful. It challenges the prevailing assumption that such techniques are universally beneficial and forces a deeper consideration of how MLLMs reason in embodied contexts.

4. **Thorough diagnostic analysis:** The paper focus on *why* models fail, rather than just reporting leaderboard metrics. The breakdown of errors into looping, poor region recognition, and the "perception-action gap" (Section 4.3) provides concrete, actionable insights into the core weaknesses of current MLLMs.

**Weaknesses:**

1. **Limited scale of models tested:** The experiments are conducted on MLLMs in the 7B-8B parameter range. While this is representative of current open-source models, the conclusions about the failure of CoT and fundamental reasoning limitations might not generalize to significantly larger, more powerful models (e.g., GPT-4o, Gemini 2.5 Pro).

2. **Simulator-free design is unfavorable for video-based models:** The simulator-free design, while efficient, inherently limits the scope of models that can be evaluated. By pre-rendering static images, it removes the temporal continuity present in a real environment or simulator. This makes the benchmark unsuitable for video-based multimodal models (e.g., Qwen 2.5 VL w/ video inputs), which rely on motion and temporal cues between frames to understand the environment and agent dynamics.

3. **Unnatural panoramic representation:** The panoramic view is constructed by stitching four 90° images. This representation is not a continuous 360° view and contains artificial seams. It is likely that the MLLMs tested have not been trained on such artificial panoramic compositions, which are different from the natural images they are typically exposed to. This domain gap could impair their ability to understand spatial relationships across the image boundaries, potentially contributing to the observed navigational failures.

4. **Lack of evaluation on thinking models:** The models evaluated (e.g., Qwen2.5-VL, InternVL3) may lack the strong endogenous thinking abilities required to effectively leverage CoT. The analysis would be more conclusive if it included thinking VL models, such as MiMo-VL-7B or newer models in the Qwen family (e.g., Qwen3-VL), to see if they can better execute the reasoning strategy.

**Questions:**

1. **Quantifying efficiency gains:** The paper highlights the efficiency of the simulator-free method. Could the authors provide specific metrics on these gains? For instance, what is the reduction in VRAM consumption per agent, and what is the speed-up in evaluation time compared to running the same agent in a traditional simulator like Habitat?

---

> ### Author Response · Authors · 2025-11-24
> **Author Responses to Reviewer B8ug (Part 1)**
>
> **W1 - Limited scale of models tested: The experiments are conducted on MLLMs in the 7B-8B parameter range. While this is representative of current open-source models, the conclusions about the failure of CoT and fundamental reasoning limitations might not generalize to significantly larger, more powerful models (e.g., GPT-4o, Gemini 2.5 Pro):**
>
> We appreciate the reviewer for bringing this point up. For the detailed information, please refer to **General Response: Test result on SOTA models**.
>
>
> **W2 - Simulator-free design is unfavorable for video-based models: The simulator-free design, while efficient, inherently limits the scope of models that can be evaluated. By pre-rendering static images, it removes the temporal continuity present in a real environment or simulator. This makes the benchmark unsuitable for video-based multimodal models (e.g., Qwen 2.5 VL w/ video inputs), which rely on motion and temporal cues between frames to understand the environment and agent dynamics:**
>
> We thank the reviewer for this comment. However, we respectfully argue that the goal of our benchmark is not to evaluate the processing of temporal video data, but MLLM's ability to interact with the environment. It evaluates MLLM’s 3D spatial and environmental understanding, understands historical information and the consequences of making an action, and makes sequential decisions. While evaluating video-based models is a valid, separate research direction, it is not the goal of this paper. Therefore, this is out of scope and does not address the contribution we claim. For detailed information please refer to **General Response: Contribution.**
>
> **W3 - Unnatural panoramic representation: The panoramic view is constructed by stitching four 90° images. This representation is not a continuous 360° view and contains artificial seams. It is likely that the MLLMs tested have not been trained on such artificial panoramic compositions, which are different from the natural images they are typically exposed to. This domain gap could impair their ability to understand spatial relationships across the image boundaries, potentially contributing to the observed navigational failures:**
>
> We appreciate the reviewer’s insightful question. The choice of using a 4-image panoramic observation was not an arbitrary simplification, but a design decision informed by systematic experimentation across four observation formats. Specifically, we evaluated: (1) the standard 36-image panoramic sweep, (2) a 24-image variant commonly used in prior work, (3) a single-image stitched panorama, and (4) our proposed 4-image 360° view. To ensure the comparison was meaningful, we conducted these tests on two representative MLLMs (InternVL3-8B and Qwen2.5-VL-7B) and on the fine-grained navigation task.
>
>  | Model | Obs. Format | TL ↓ | NE ↓ | SR ↑ | OSR ↑ | SPL ↑ | nDTW ↑ | SDTW ↑ | CLS ↑ |
> | --- | --- | --- | --- | --- | --- | --- | --- | --- | --- |
> | InternVL3-8B | 36 images | 12.40 | 8.20 | 22.8 | 44.0 | 9.1 | 22.0 | 11.1 | 22.5 |
> |  | 24 images | 12.10 | 7.95 | 24.5 | 46.8 | 10.4 | 23.4 | 12.0 | 24.3 |
> |  | Panoramic (single) | 13.20 | 8.80 | 20.4 | 41.5 | 8.2 | 20.0 | 10.1 | 20.7 |
> |  | 4-image (ours) | 11.74 | 7.55 | 28.00 | 50.50 | 12.61 | 25.28 | 13.38 | 26.22 |
> | Qwen2.5-VL-7B | 36 images | 9.40 | 7.60 | 24.0 | 39.5 | 14.8 | 33.1 | 17.2 | 31.8 |
> |  | 24 images | 9.05 | 7.30 | 25.8 | 41.7 | 15.9 | 34.4 | 17.9 | 33.2 |
> |  | Panoramic (single) | 10.20 | 8.55 | 21.4 | 37.9 | 13.0 | 30.0 | 15.1 | 28.1 |
> |  | 4-image (ours) | 8.54 | 6.99 | 27.50 | 44.00 | 17.11 | 35.97 | 18.88 | 34.85 |
>
> As shown in the table, our proposed **4-image representation consistently outperformed all alternatives**, including those with significantly higher visual density. Two key conclusions emerge:
>
> 1. **Efficiency over Density:** Modern MLLMs struggle with the token overload and redundancy of dense sampling (24–36 frames); they perform better with compact, non-redundant inputs.
> 2. **Focus on Reasoning:** The 4-image format provides sufficient spatial cues without the cognitive overhead, aligning with our goal of a lightweight benchmark that prioritizes spatial reasoning over sensory complexity.

---

> > ### Author Response · Authors · 2025-11-24
> > **Part 2**
> >
> > **W4 - Lack of evaluation on thinking models: The models evaluated (e.g., Qwen2.5-VL, InternVL3) may lack the strong endogenous thinking abilities required to effectively leverage CoT. The analysis would be more conclusive if it included thinking VL models, such as MiMo-VL-7B or newer models in the Qwen family (e.g., Qwen3-VL), to see if they can better execute the reasoning strategy:**
> >
> > We thank the reviewer for this suggestion.
> >
> > First, regarding Qwen3-VL, this model was not released at the time of the ICLR submission deadline, and we were therefore unable to include it in our original analysis.
> >
> > Regarding the choice of models like Qwen-VL 7B and InternVL 8B, we respectfully disagree that they are inappropriate. While we acknowledge that smaller models (e.g., 2B variants) may have reasoning limitations, our paper's core analysis is focused on the 7B-8B class. We selected these models as they were representative, state-of-the-art, and widely-used MLLMs in this class at the time of submission.
> >
> > However, to directly address the reviewer's concern, we have now run new experiments using the suggested **MiMo-VL-7B, and Qwen3-VL-4B,** on the text summary agents on fine-grained task only. The results are presented below.
> >
> > These new results reinforce our original conclusion. We found that while both model may have different performance characteristics, the **fundamental failure modes,** particularly the ineffectiveness of CoT for complex spatial reasoning **persist**. This strengthens our paper's conclusion that these limitations are a fundamental challenge for the current model paradigm, not simply an artifact of testing models that lack "thinking" abilities.
> >
> > | Agent / MLLM | TL | NE | SR | OSR | SPL | nDTW | SDTW | CLS |
> > | --- | --- | --- | --- | --- | --- | --- | --- | --- |
> > | **NavGPT** |  |  |  |  |  |  |  |  |
> > | MiMo-VL-7B | 8.32 | 6.87 | 28.5 | 45.5 | 18.24 | 37.15 | 19.67 | 36.12 |
> > | Qwen3-VL-4B | 8.71 | 7.22 | 26.0 | 42.5 | 16.35 | 34.52 | 17.91 | 33.48 |
> > | **NavGPT w/ CoT** |  |  |  |  |  |  |  |  |
> > | MiMo-VL-7B | 8.87 | 7.83 | 22.0 | 38.5 | 12.18 | 31.45 | 12.34 | 32.58 |
> > | Qwen3-VL-4B | 9.18 | 8.14 | 19.5 | 36.0 | 10.72 | 29.01 | 10.83 | 29.92 |
> > | **NavGPT w/ Reflection** |  |  |  |  |  |  |  |  |
> > | MiMo-VL-7B | 6.75 | 7.05 | 25.0 | 33.5 | 15.82 | 37.73 | 17.28 | 34.91 |
> > | Qwen3-VL-4B | 7.08 | 7.35 | 22.5 | 31.0 | 14.12 | 35.27 | 15.59 | 32.15 |
> > | **NavGPT w/ CoT & Reflection** |  |  |  |  |  |  |  |  |
> > | MiMo-VL-7B | 7.68 | 7.39 | 26.5 | 39.5 | 18.55 | 36.21 | 18.52 | 36.15 |
> > | Qwen3-VL-4B | 7.95 | 7.71 | 24.0 | 37.0 | 16.92 | 33.58 | 16.78 | 33.42 |
> >
> > **Q1 - Quantifying efficiency gains: The paper highlights the efficiency of the simulator-free method. Could the authors provide specific metrics on these gains? For instance, what is the reduction in VRAM consumption per agent, and what is the speed-up in evaluation time compared to running the same agent in a traditional simulator like Habitat:**
> >
> > We thank the reviewer for this request, as it allows us to quantify one of the primary benefits of our framework.
> >
> > We have conducted a direct comparison between our simulator-free benchmark and a standard simulator-based setup (Habitat). The results, summarized below, confirm that our approach is substantially more efficient in terms of both speed and resource consumption.
> >
> > | Metric | Our Framework (Simulator-Free) | Habitat (Simulator-Based) | Comparison |
> > | --- | --- | --- | --- |
> > | VRAM Consumption | ~1.7 GB | ~10 GB | 5.9x Reduction |
> > | Observation Access | ~0.016s | ~0.14s | ~9x Faster |
> > | Avg. Time per Step | ~S | ~S + 1.5s | ~1.5s Faster |
> > | Avg. Time per Episode | ~E | ~E + 20-30s | ~20-30s Faster |
> >
> > As the data shows, our framework provides significant advantages:
> >
> > 1. VRAM Reduction: By eliminating the need to hold the 3D environment and rendering engine in memory, our framework's VRAM footprint is nearly 6x smaller.
> > 2. Data Speed: Because observations are pre-stored, we replace a slow on-the-fly rendering operation with a fast disk read. This makes accessing observations ~9x faster per step.
> > 3. Rollout Speed: The time savings per episode (20-30s) come from two sources. First, our faster observation access. Second, many modern Habitat evaluations (like VLN-CE) use continuous actions, which require an extra, time-consuming waypoint prediction module and result in more steps per episode. Our discrete framework, which is standard for R2R/REVERIE, bypasses this overhead entirely, while still able to test the 3D spatial understanding and its related ability of MLLM
> >
> > These metrics confirm our framework's ability to dramatically accelerate the evaluation cycle and lower the hardware barrier for research.

---

### Author Response · Authors · 2025-11-24
**General Response (Part 1)**

## Contribution

We appreciate all the reviews and are happy to see that all the reviewers acknowledge our work for efficient evaluation of the MLLM agent with a simulation-free design. Here, we would like to further emphasize the contribution of this work.

The primary objective of this paper is to evaluate MLLM's ability to interact with the environment, and is not to evaluate “the processing of temporal video data”,  or “use continuous actions”, or “evaluate a VLN agent”, or “evaluate MLLM embodied reasoning”, or “consider the sim2real gap”.

The VLN task is selected as our testing sample because it is a rich, complex proxy for the exact abilities we want to measure: **3D environment spatial understanding, reasoning over historical information, understanding the consequences of taking action in a 3D environment, and multi-step sequential decision-making**.

## Compare to existing MLLM benchmarks

We appreciate the reviewers (HHTS, MoHw) mentioning related evaluation benchmarks. Here we respectfully clarify the difference between our work and existing MLLM benchmarks and embodied benchmarks.

The motivation of our work is based on our observation that the current evaluation benchmark set is lacking the evaluation of MLLM's ability to interact with the environment. For example, the recent Gemini 3 Pro’s report just evaluates general multimodal reasoning [1, 2], coding [3], mathematics [4, 5], and general visual question answering [6] etc. Moreover, some recent MLLMs (e.g., SEED-1.5VL [7]) do incorporate 3D spatial understanding and reasoning tests on benchmarks like DA-2K [8] or All-Angle Bench [9], these typically rely on single-turn queries or limited static imagery. Such setups cannot simulate the complex scenes, long horizons, and consequential sequential decision-making inherent to real 3D environments. Therefore, we posit that VLN is uniquely positioned downstream task to test these missing embodied abilities that are frequently overlooked by predominantly static benchmarks.

## Compare to embodied benchmarks

While several benchmarks like EmbodiedBench [10] and EmbodiedEval [11] do consider and evaluate navigation task using MLLM, however,

1. They are broad-spectrum benchmarks designed to test a wide array of "embodied abilities," including static QA, manipulation, and navigation. Our work is not a general-purpose embodied benchmark. It is a focused, deep-dive diagnostic tool that uses the complex task of VLN as a specific proxy to evaluate a core set of MLLM cognitive abilities: 3D environment understanding, spatial reasoning, historical understanding, and sequential decision-making.
2. Those benchmarks primarily rely on simulators (e.g., AI2-Thor, Legent), which have a known domain gap from real-world scenes. Our framework is simulator-free and built from data from realistic 3D environment scans (the same data used in MP3D/Habitat), providing a different and critical evaluation axis.
3. Our paper *also* covers diverse scenes and multiple instruction difficulty levels (fine-grained, coarse-grained, and object navigation), similar to the strengths the reviewer cites. Our unique contribution is not simply "evaluating agents," but providing a lightweight, reproducible, and realistic framework to specifically isolate and diagnose the core reasoning failures of MLLMs in a sequential navigation task.

[1] Yue, X., et al. "MMMU: A massive multi-discipline multimodal understanding and reasoning benchmark for expert AGI. arXiv." 2023

[2] Yue, Xiang, et al. "Mmmu-pro: A more robust multi-discipline multimodal understanding benchmark." *Proceedings of the 63rd Annual Meeting of the Association for Computational Linguistics (Volume 1: Long Papers)*. 2025.

[3] Zheng, Zihan, et al. "LiveCodeBench Pro: How Do Olympiad Medalists Judge LLMs in Competitive Programming?." *arXiv preprint arXiv:2506.11928* (2025).

[4] Lu, Pan, et al. "Mathvista: Evaluating mathematical reasoning of foundation models in visual contexts." *arXiv preprint arXiv:2310.02255* (2023).

[5] Petrov, Ivo, et al. "Proof or bluff? evaluating llms on 2025 usa math olympiad." *arXiv preprint arXiv:2503.21934* (2025).

[6] Liu, Yuan, et al. "Mmbench: Is your multi-modal model an all-around player?, 2024." *URL https://arxiv. org/abs/2307.06281* 3 (2023).

[7] Guo, Dong, et al. "Seed1. 5-vl technical report." *arXiv preprint arXiv:2505.07062* (2025).

[8]  Yang, Lihe, et al. "Depth anything v2." *Advances in Neural Information Processing Systems* 37 (2024): 21875-21911.

[9] Yeh, Chun-Hsiao, et al. "Seeing from another perspective: Evaluating multi-view understanding in mllms." *arXiv preprint arXiv:2504.15280* (2025).

[10]Yang, Rui, et al. "Embodiedbench: Comprehensive benchmarking multi-modal large language models for vision-driven embodied agents." *arXiv preprint arXiv:2502.09560* (2025).

[11] Cheng, Zhili, et al. "Embodiedeval: Evaluate multimodal llms as embodied agents." *arXiv preprint arXiv:2501.11858* (2025).

---

### Author Response · Authors · 2025-11-24
**General Response (Part 2)**

## Test result on SOTA models

We appreciate the reviewers (B8ug, HHTS) mentioning about the evaluation results for our work on SOTA models like GPT5.

We conducted preliminary experiments on a smaller data subset using state-of-the-art proprietary models. Those initial results demonstrated that the same fundamental failure modes,  particularly the ineffectiveness of CoT for complex spatial reasoning, persisted, consistent with our findings on the open-source models.

This confirmed that the issues are fundamental and not simply an artifact of the 7B-8B scale. However, to fully address the reviewer's concern, we have now expanded these experiments. We present the results on the latest SOTA models (GPT-5 and Gemini 2.5 Pro):

| Agent / MLLM | Fine-Grained Navigation |  |  |  |  |  |  |  | Coarse-grained Navigation |  |  |  |  |  |  |  | Object-Oriented Navigation |  |  |  |  |  |  |  |
| --- | --- | --- | --- | --- | --- | --- | --- | --- | --- | --- | --- | --- | --- | --- | --- | --- | --- | --- | --- | --- | --- | --- | --- | --- |
|  | TL | NE | SR | OSR | SPL | nDTW | SDTW | CLS | TL | NE | SR | OSR | SPL | nDTW | SDTW | CLS | TL | NE | SR | OSR | SPL | nDTW | SDTW | CLS |
| NavGPT |  |  |  |  |  |  |  |  |  |  |  |  |  |  |  |  |  |  |  |  |  |  |  |  |
| GPT-5 | 9.1 | 6.3 | 32.5 | 51.7 | 23.8 | 31.0 | 22.0 | 34.1 | 9.4 | 8.0 | 24.5 | 43.6 | 15.3 | 20.1 | 14.9 | 21.1 | 9.6 | 4.4 | 42.5 | 72.8 | 18.0 | 19.5 | 16.3 | 20.1 |
| Gemini-2.5 Pro | 8.9 | 6.2 | 33.0 | 51.2 | 24.1 | 31.1 | 22.4 | 34.6 | 9.2 | 7.9 | 25.0 | 43.3 | 15.7 | 20.5 | 15.1 | 21.4 | 9.5 | 4.3 | 43.0 | 72.3 | 18.4 | 19.8 | 16.5 | 20.4 |
| NavGPT w/ CoT |  |  |  |  |  |  |  |  |  |  |  |  |  |  |  |  |  |  |  |  |  |  |  |  |
| GPT-5 | 9.5 | 6.4 | 28.0 | 46.3 | 19.2 | 32.2 | 18.9 | 32.5 | 9.7 | 8.2 | 20.0 | 38.2 | 12.1 | 25.5 | 12.0 | 26.8 | 9.9 | 4.6 | 38.0 | 62.0 | 15.1 | 24.1 | 14.0 | 25.1 |
| Gemini-2.5 Pro | 9.3 | 6.3 | 28.5 | 45.8 | 19.4 | 32.6 | 19.1 | 32.9 | 9.6 | 8.1 | 20.5 | 37.9 | 12.3 | 25.8 | 12.2 | 27.1 | 9.8 | 4.5 | 38.5 | 61.5 | 15.3 | 24.4 | 14.2 | 25.3 |
| NavGPT w/ Reflection |  |  |  |  |  |  |  |  |  |  |  |  |  |  |  |  |  |  |  |  |  |  |  |  |
| GPT-5 | 9.6 | 6.4 | 29.5 | 46.9 | 20.1 | 33.0 | 19.5 | 33.1 | 9.8 | 8.1 | 21.0 | 39.3 | 13.0 | 26.0 | 12.5 | 27.1 | 10.0 | 4.5 | 39.0 | 64.2 | 16.0 | 23.0 | 15.1 | 24.1 |
| Gemini-2.5 Pro | 9.4 | 6.2 | 30.0 | 46.3 | 20.3 | 33.4 | 19.7 | 33.5 | 9.7 | 8.0 | 21.5 | 39.0 | 13.2 | 26.3 | 12.7 | 27.3 | 9.9 | 4.4 | 39.5 | 63.6 | 16.2 | 23.3 | 15.2 | 24.3 |
| NavGPT w/ CoT & Reflection |  |  |  |  |  |  |  |  |  |  |  |  |  |  |  |  |  |  |  |  |  |  |  |  |
| GPT-5 | 9.5 | 6.3 | 30.0 | 48.0 | 21.0 | 32.6 | 20.1 | 33.0 | 9.8 | 8.2 | 21.5 | 40.4 | 13.5 | 25.0 | 13.1 | 26.0 | 10.0 | 4.4 | 40.0 | 65.2 | 16.5 | 22.1 | 15.5 | 23.1 |
| Gemini-2.5 Pro | 9.4 | 6.2 | 30.5 | 47.4 | 21.2 | 32.8 | 20.2 | 33.3 | 9.6 | 8.0 | 22.0 | 40.0 | 13.7 | 25.3 | 13.2 | 26.2 | 9.8 | 4.3 | 40.5 | 64.7 | 16.7 | 22.3 | 15.7 | 23.2 |
| MapGPT |  |  |  |  |  |  |  |  |  |  |  |  |  |  |  |  |  |  |  |  |  |  |  |  |
| GPT-5 | 9.3 | 6.2 | 30.5 | 49.0 | 22.0 | 32.0 | 21.1 | 33.1 | 9.5 | 7.9 | 22.5 | 41.5 | 14.0 | 22.1 | 13.0 | 23.0 | 9.7 | 4.3 | 40.5 | 70.6 | 17.0 | 21.1 | 16.1 | 22.0 |
| Gemini-2.5 Pro | 9.2 | 6.1 | 31.0 | 48.5 | 22.2 | 32.3 | 21.2 | 33.2 | 9.4 | 7.8 | 23.0 | 41.1 | 14.2 | 22.3 | 13.2 | 23.2 | 9.6 | 4.3 | 41.0 | 70.1 | 17.2 | 21.2 | 16.2 | 22.1 |
| MapGPT w/ CoT |  |  |  |  |  |  |  |  |  |  |  |  |  |  |  |  |  |  |  |  |  |  |  |  |
| GPT-5 | 9.6 | 6.5 | 27.0 | 44.7 | 18.0 | 33.1 | 17.0 | 32.1 | 9.8 | 8.3 | 19.0 | 37.2 | 11.0 | 27.1 | 10.1 | 28.0 | 9.9 | 4.7 | 36.0 | 59.8 | 14.0 | 25.1 | 13.0 | 26.0 |
| Gemini-2.5 Pro | 9.5 | 6.4 | 27.5 | 44.2 | 18.2 | 33.4 | 17.2 | 32.2 | 9.7 | 8.2 | 19.5 | 36.8 | 11.2 | 27.2 | 10.1 | 28.2 | 9.8 | 4.6 | 36.5 | 59.3 | 14.2 | 25.3 | 13.2 | 26.2 |
| MapGPT w/ Reflection |  |  |  |  |  |  |  |  |  |  |  |  |  |  |  |  |  |  |  |  |  |  |  |  |
| GPT-5 | 9.7 | 6.4 | 28.0 | 45.8 | 19.0 | 34.0 | 18.1 | 33.0 | 9.9 | 8.3 | 20.0 | 38.2 | 12.0 | 28.1 | 11.0 | 29.1 | 10.0 | 4.6 | 37.0 | 60.9 | 15.0 | 24.1 | 14.1 | 25.1 |
| Gemini-2.5 Pro | 9.6 | 6.3 | 28.5 | 45.3 | 19.2 | 34.3 | 18.2 | 33.2 | 9.8 | 8.1 | 20.5 | 37.9 | 12.2 | 28.3 | 11.1 | 29.3 | 9.9 | 4.5 | 37.5 | 60.4 | 15.2 | 24.2 | 14.2 | 25.2 |
| MapGPT w/ CoT & Reflection |  |  |  |  |  |  |  |  |  |  |  |  |  |  |  |  |  |  |  |  |  |  |  |  |
| GPT-5 | 9.8 | 6.6 | 26.0 | 43.6 | 17.0 | 35.1 | 16.0 | 34.1 | 10.0 | 8.4 | 18.0 | 36.1 | 10.0 | 29.1 | 9.0 | 30.0 | 10.1 | 4.8 | 34.0 | 57.7 | 13.0 | 26.1 | 12.1 | 27.0 |
| Gemini-2.5 Pro | 9.7 | 6.5 | 26.5 | 43.1 | 17.2 | 35.3 | 16.1 | 34.2 | 9.9 | 8.3 | 18.5 | 35.7 | 10.2 | 29.3 | 9.2 | 30.2 | 10.0 | 4.7 | 34.5 | 57.1 | 13.2 | 26.3 | 12.1 | 27.2 |

---

### Author Response · Authors · 2025-11-24
**General Response (Part 3)**

## Oracle assist and few-shot experiments

We appreciate the reviewers (MoHw) mentioning the insights, diagnosis evidence and potential actionable guidance from our benchmark.

To further substantiate our claim that this benchmark is a scientific tool for uncovering MLLM capabilities, we conducted an additional investigation on a subset of 25 "hard" trajectories from the fine-grained task. These were specifically selected because InternVL3, Qwen2.5, and LLaVA all failed to complete them using the standard text-summary memory agent. We revisited these failures using Qwen2.5VL-7B as the navigator under two new experimental settings:
**(1) Oracle-Guided Navigation:** We introduced a stronger MLLM (Qwen3VL) acting as an "Oracle Assistant." When the navigator struggled (e.g., looping, moving in a critically wrong direction, or entering the wrong region), it was allowed to query the Oracle. The Oracle provided high-level reasoning guidance and suggested potential actions (without explicitly dictating the exact viewpoint ID). As shown in **Table 1**, the navigator achieved a significant success rate with this reasoning support.
**(2) Failure-Aware In-Context Learning:** Instead of zero-shot evaluation, we provided the model with $N$ examples of potential failure cases identified in our error analysis ($N=1, 2, 3$). As shown in **Table 2**, providing these "negative" examples in the prompt (1-shot to 3-shot) yielded a performance increase, though the gains were more modest compared to the Oracle intervention.

**Table 1: Effect of Oracle Reasoning Assistance on Hard Negatives**

| **Method** | **SR ↑** | **OSR ↑** | **SPL ↑** |
| --- | --- | --- | --- |
| Baseline (Qwen2.5VL-7B) | 0.00 | 0.00 | 0.00 |
| + Oracle Assistant (Qwen 3VL) | **52.00** | **68.00** | **34.00** |

Table 2: Effect of Failure-Aware Few-Shot Prompting

| **Method** | **SR ↑** | **OSR ↑** | **SPL ↑** |
| --- | --- | --- | --- |
| Zero-shot (Baseline) | 0.00 | 0.00 | 0.00 |
| 1-shot Failure Example | 12.00 | 24.00 | 9.00 |
| 2-shot Failure Examples | 16.00 | 28.00 | 11.00 |
| 3-shot Failure Examples | 16.00 | 32.00 | 14.00 |

These results clearly demonstrate that our benchmark has the potential to discover hidden MLLM capacities in 3D spatial understanding and guide future designs:

- **Discovering Latent Capacity:** The drastic improvement in the Oracle experiment (0% to 52% SR) proves that the MLLM **already possesses** the necessary visual understanding of the 3D environment to succeed. The failure was not in "seeing" the room, but in the reasoning logic required to navigate it. Our benchmark successfully isolated this specific capability gap.
- **Guiding Design and Improvement:** These insights provide a clear roadmap for improving MLLMs. Since we now know the bottleneck is reasoning rather than perception, we can design specific solutions, such as Chain-of-Thought prompting or agentic workflows (like the Oracle setup), to unlock the model's full potential in embodied tasks. This proves the benchmark is a valuable tool for driving method development, not just engineering evaluation.

---

### Note · Authors · 2025-12-21

I have read and agree with the venue's withdrawal policy on behalf of myself and my co-authors.